# Dendritic cell Piezo1 directs the differentiation of $T_H1$ and $T_{reg}$ cells in cancer

Yuexin Wang[1†], Hui Yang[2†], Anna Jia[1†], Yufei Wang[1†], Qiuli Yang[1†], Yingjie Dong[1], Yueru Hou[1], Yejin Cao[1], Lin Dong[1], Yujing Bi[3*], Guangwei Liu[1*]

[1]Key Laboratory of Cell Proliferation and Regulation Biology, Ministry of Education, College of Life Sciences, Beijing Normal University, Beijing, China; [2]Department of Immunology, School of Basic Medical Sciences, Fudan University, Shanghai, China; [3]State Key Laboratory of Pathogen and Biosecurity, Beijing Institute of Microbiology and Epidemiology, Beijing, China

**Abstract** Dendritic cells (DCs) play an important role in anti-tumor immunity by inducing T cell differentiation. Herein, we found that the DC mechanical sensor Piezo1 stimulated by mechanical stiffness or inflammatory signals directs the reciprocal differentiation of $T_H1$ and regulatory T ($T_{reg}$) cells in cancer. Genetic deletion of Piezo1 in DCs inhibited the generation of $T_H1$ cells while driving the development of $T_{reg}$ cells in promoting cancer growth in mice. Mechanistically, Piezo1-deficient DCs regulated the secretion of the polarizing cytokines TGFβ1 and IL-12, leading to increased TGFβR2-p-Smad3 activity and decreased IL-12Rβ2-p-STAT4 activity while inducing the reciprocal differentiation of $T_{reg}$ and $T_H1$ cells. In addition, Piezo1 integrated the SIRT1-hypoxia-inducible factor-1 alpha (HIF1α)-dependent metabolic pathway and calcium-calcineurin-NFAT signaling pathway to orchestrate reciprocal $T_H1$ and $T_{reg}$ lineage commitment through DC-derived IL-12 and TGFβ1. Our studies provide critical insight for understanding the role of the DC-based mechanical regulation of immunopathology in directing T cell lineage commitment in tumor microenvironments.

**\*For correspondence:**
byj7801@sina.com (YB);
liugw@bnu.edu.cn (GL)

[†]These authors contributed equally to this work

**Competing interest:** The authors declare that no competing interests exist.

## Editor's evaluation

In the present study, the authors use mouse cancer models to study the role of Piezo1 on DC-mediated priming of CD4+ T cells. They show that Piezo1 knockout results in faster tumor progression and accumulation of more regulatory T cells, and that Smad3 and STAT4 are involved in DC-mediated differentiation of $T_H1$ and $T_{reg}$ cells. Overall this represents a mechanistic advance in our understanding of DC biology as it relates to cancer. This also has the potential to extend beyond cancer to better our understanding of DC-mediated T cell differentiation.

## Introduction

CD4+ T helper cells play a central role in cancer by differentiating into different T cell subsets including $T_H1$ cells, $T_H2$ cells, $T_H9$ cells, $T_H17$ cells, and regulatory T ($T_{reg}$) cells, in specific cytokine environments (*Zhu et al., 2010*; *Adorini, 2003*). Foreign antigens and all kinds of innate stimuli are often presented by antigen-presenting cells (APCs) to further direct the development and differentiation of different subsets of CD4+ T cells in tumor (*Steinman et al., 2003*; *Marin et al., 2019*). Inflammatory stimuli, such as bacterial lipopolysaccharide (LPS), cytokine, or other innate stimuli, such as oxygen, nutrient availability, and even force and pressure, can alter the immune responses. In particular, the tumor microenvironment often integrates different innate physiological or pathological stimuli to develop a

complex stimulation microenvironment (*Solis et al., 2019*; *Geng et al., 2021*). As professional APCs, dendritic cells (DCs) respond to various exogenous and endogenous stimuli mainly through three key signal pathways, including costimulatory molecule expression, T cell receptor (TCR) signaling, and cytokine production, bridging innate and adaptive immunity, and regulating the differentiation of different T cell subsets (*Zhu et al., 2010*; *Murphy and Stockinger, 2010*; *Zhou et al., 2008*; *Chi, 2012*; *McKinstry et al., 2010*; *Li and Flavell, 2008*; *Stockinger et al., 2007*; *Zhu and Paul, 2010*). DC-derived cytokines and chemokines exert proinflammatory or anti-inflammatory effects and are involved in the shaping of distinct T cell subset lineage programs to determine the prognosis of cancer patients. But, how the differentiation of CD4+ T cells is modulated and regulated by innate immune signaling pathways in DCs in the tumor microenvironment remains unclear.

Piezo1 was originally identified as a mechanically activated non-selective cation ion channel with significant permeability to calcium ions, is evolutionary conserved and is involved in the proliferation and development of various types of cells in the context of various types of mechanical or innate stimuli. It has been reported that innate inflammatory stimulation or mechanical changes, such as changes in stiffness, can activate Piezo1 and trigger an inflammatory response (*Geng et al., 2021*; *Yang et al., 2008*). Piezo1 exerts significant regulatory effects on many kinds of immune cell functions including macrophages, DCs, and T cells in inflammation and cancer (*Solis et al., 2019*; *Geng et al., 2021*; *Aykut et al., 2020*; *Atcha et al., 2021*; *Chakraborty et al., 2021*; *Jairaman et al., 2021*). However, it is still unclear whether Piezo1-targeted DCs affect the differentiation of different subsets of T cells in cancer.

Herein, we found that the DC mechanical sensor Piezo1 stimulated by mechanical stiffness or inflammatory signals directs the reciprocal differentiation of $T_H1$ and $T_{reg}$ cells in inhibiting tumor growth.

## Results

### Inflammatory and stiffness stimuli alter Piezo1 expressions by DCs

As reported (*Chakraborty et al., 2021*; *Liu et al., 2018*), environmental stiffness can alter the secretion of inflammatory factors by DCs and the mechanical force receptor Piezo1 may be involved in this regulation. We first examined the effect of environmental stiffness on the expression of Piezo1 in DCs. We used a cell culture system consisting of a polydimethylsiloxane (PDMS) hydrogel-coated plate, as described (*Yang et al., 2008*; *Liu et al., 2018*). The mechanical properties of the PDMS hydrogel can be changed by adjusting the matrix/curing agent ratio, and this ratio can be precisely adjusted to simulate physiological tension. Consistent with previous reports (*Yang et al., 2008*; *Liu et al., 2018*), we used 2 kPa to mimic lymphoid tissue under physiological conditions and 50 kPa to mimic lymphoid tissue under inflammatory conditions. Sorted splenic DCs cultured on a stiff hydrogel (E=50 kPa) or plastic plates exhibited significantly enhanced expression of the proinflammatory cytokine IL-12 and diminished expression of the anti-inflammatory cytokine TGFβ1 compared with DCs cultured on a pliant hydrogel (E=2 kPa; *Figure 1—figure supplement 1A*). This suggests that substrate stiffness could regulate inflammatory cytokine secretion by DCs. Moreover, the expression of the mechanical force receptor Piezo1 was significantly upregulated in 50 kPa-conditioned hydrogels compared with 2 kPa-conditioned hydrogels, similar to that in DCs stimulated with the inflammatory stimulus LPS (*Figure 1—figure supplement 1B*). Although 50 kPa-conditioned hydrogels significantly caused more IL-12 and less TGFβ1 compared with 2 kPa, mechanical force receptor Piezo1-deficient DC cells significant rescue them to normal level (*Figure 1—figure supplement 1C*). These data suggested that mechanical force receptor Piezo1 mediated the inflammatory cytokine production induced by substrate stiffness in DCs.

### DC-specific Piezo1-deficient mice exhibited altered T cell differentiation

To investigate the regulatory effect of Piezo1 expressed by DCs on T cell function, we generated DC-specific Piezo1 conditional knockout (*Piezo1-/-*) mice with *Piezo1*flox/flox and *Cd11c-cre. Piezo1-/-* mice showed no obvious abnormalities after birth. However, after 40 weeks of age, these mice showed lower weight loss and less T cell activation in mesenteric lymph nodes (MLNs), Peyer's patches (PPs), intraepithelial lymphocytes (IELs), and lamina propria lymphocytes (LPLs) than WT mice (*Figure 1— figure supplement 2*). Importantly, Piezo1 deficiency in DCs results in fewer IFNγ+ $T_H1$ cells and more

Foxp3$^+$ $T_{reg}$ cells but does not affect the numbers of $T_H$2 or $T_H$17 cells (**Figure 1—figure supplement 3**). Thus, we conclude that DC-specific Piezo1 deficiency alters the differentiation of $T_H$1 and $T_{reg}$ cells in aged mice, which might be related to clinical manifestations.

## DC-specific Piezo1 regulates T cell differentiation in promoting tumor growth

Next, we studied the effect of DC-specific Piezo1 deletion on T cell differentiation in MC38 mouse colon cancer. We observed changes in tumor growth in *Piezo1$^{-/-}$* and WT mice. The rate of tumor growth was significantly faster and greater in *Piezo1$^{-/-}$* than in WT mice (**Figure 1A**). *Piezo1$^{-/-}$* mice had more Foxp3$^+$ $T_{reg}$ cells, fewer IFNγ$^+$ $T_H$1 cells, and normal numbers of $T_H$2, $T_H$17, CD8$^+$T cell, and IFNγ$^+$CD8$^+$T cells in tumor tissue compared with WT control (**Figure 1B**, **Figure 1—figure supplement 4A-B**). Furthermore, we isolated T cells from draining lymph nodes (dLNs) of tumor-bearing mice at day 20, 30, and 40 and observed dynamic T cell differentiation. Both *Tbx21* and *Ifng* levels were downregulated rapidly, *Foxp3* expression in T cells was upregulated gradually, and *Il4*, *Il10*, *Il17a*, and *Gata3*, *Rorgt* expressions did not change in *Piezo1$^{-/-}$* mice (**Figure 1C**, **Figure 1—figure supplement 4C**). Similar tumor growth and T cell differentiation were also observed in B16.F10 melanoma tumor (**Figure 1—figure supplement 5**). Thus, these data suggest that DC-specific Piezo1 deficiency probably directs $T_{reg}$ and $T_H$1 cell differentiation to promote the tumor growth in the context of tumor microenvironment.

Further, we observed the antigen-specific responses of T cells in tumor-bearing mice during tumor growth. MC38-OVA tumor cells were implanted subcutaneously in WT and *Piezo1$^{-/-}$* mice. And at day 20 after tumor cell implantation, naïve CD45.1$^+$ OTII T cells were isolated from OTII-TCR transgenic mice, labeled with CFSE and transferred into WT and *Piezo1$^{-/-}$*-bearing tumor recipient mice (**Figure 1D**). After 10 days of adoptive transfer of OTII T cells to recipient tumor-bearing mice, the intracellular staining of the CD45.1$^+$ CFSE$^+$ donor T cells from tumors in recipient WT and *Piezo1$^{-/-}$* mice analyzed showed that the percentage of CFSE$^+$ T cells is similar between WT and *Piezo1$^{-/-}$*-bearing tumor mice. However, there are more Foxp3$^+$$T_{reg}$ cells and less IFNγ$^+$$T_H$1 cells, but similar $T_H$2 and $T_H$17 cells in *Piezo1$^{-/-}$* compared with WT bearing-tumor mice (**Figure 1D–E**). These data suggest that Piezo1 in DCs regulates the $T_H$1 and $T_{reg}$ differentiation in tumor microenvironment with the antigen-specific manner.

## DC-specific Piezo1 expression instructs antigen-specific $T_H$1 and $T_{reg}$ cell differentiation

Next, we conducted an adoptive transfer experiment to investigate the T cell response induced by DC Piezo1. Naïve T cells (CD45.2$^+$CD4$^+$TCR$^+$CD44$^{low}$CD62L$^{high}$) from C57BL/6 mice were transferred into CD45.1$^+$C57BL/6 recipient mice, and then, the recipient mice were immunized with WT and *Piezo1$^{-/-}$* splenic DCs and LPS. The donor cells were analyzed on day 10 after immunization. T cell proliferation was comparable between the WT and *Piezo1$^{-/-}$* mice (**Figure 2A**). However, the recipient mice immunized with *Piezo1$^{-/-}$* splenic DCs exhibited more Foxp3$^+$CD4$^+$ T cells and less IFNγ$^+$CD4$^+$ T cells. Both the WT and *Piezo1$^{-/-}$* mice showed similar levels of IL-4 and IL-17 expression among donor CD4$^+$ T cells (**Figure 2B–C**). Furthermore, the antigen-specific responses of donor T cells were investigated. Naïve OTII T cells were isolated and transferred into CD45.1$^+$ C57BL/6 recipient mice which immunized with WT and *Piezo1$^{-/-}$* DCs and antigen. Although WT and *Piezo1$^{-/-}$* showed similar T cell proliferation (**Figure 2D**), donor cells immunized with *Piezo1$^{-/-}$* DCs exhibited more Foxp3$^+$ cells, fewer IFNγ$^+$ cells, and similar numbers of IL-4$^+$$T_H$2 cells and IL-17A$^+$$T_H$17 cells (**Figure 2E–F**). Further, we examined DCs conditioned by 2 or 50 kPa hydrogel in adoptive transfer experiments. Splenic DCs were isolated from WT and *Piezo1$^{-/-}$* mice and plated on 2 or 50 kPa hydrogels. Naïve T cells or OTII T cells were isolated and transferred into CD45.1$^+$ recipient mice, the recipient mice were immunized with 2 or 50 kPa hydrogel-conditioned DCs with or without antigen. Donor T cells from mice immunized with *Piezo1$^{-/-}$* DCs conditioned with 50 kPa, but not 2 kPa, hydrogels included more Foxp3$^+$ $T_{reg}$ cells and less IFNγ$^+$ $T_H$1 cells (**Figure 2G–H**). Altogether, these data suggest that the mechanical sensor Piezo1 in DCs stimulated by innate inflammatory or stiffness stimuli directs the reciprocal differentiation of $T_H$1 and $T_{reg}$ cells.

## Piezo1 is required for DC-dependent T cell differentiation

Next, we investigated the effects of Piezo1 expression by DCs on T cell subset differentiation in an in vitro system. Polyclonal T cells or antigen-specific OTII T cells cocultured with *Piezo1$^{-/-}$* DCs in the

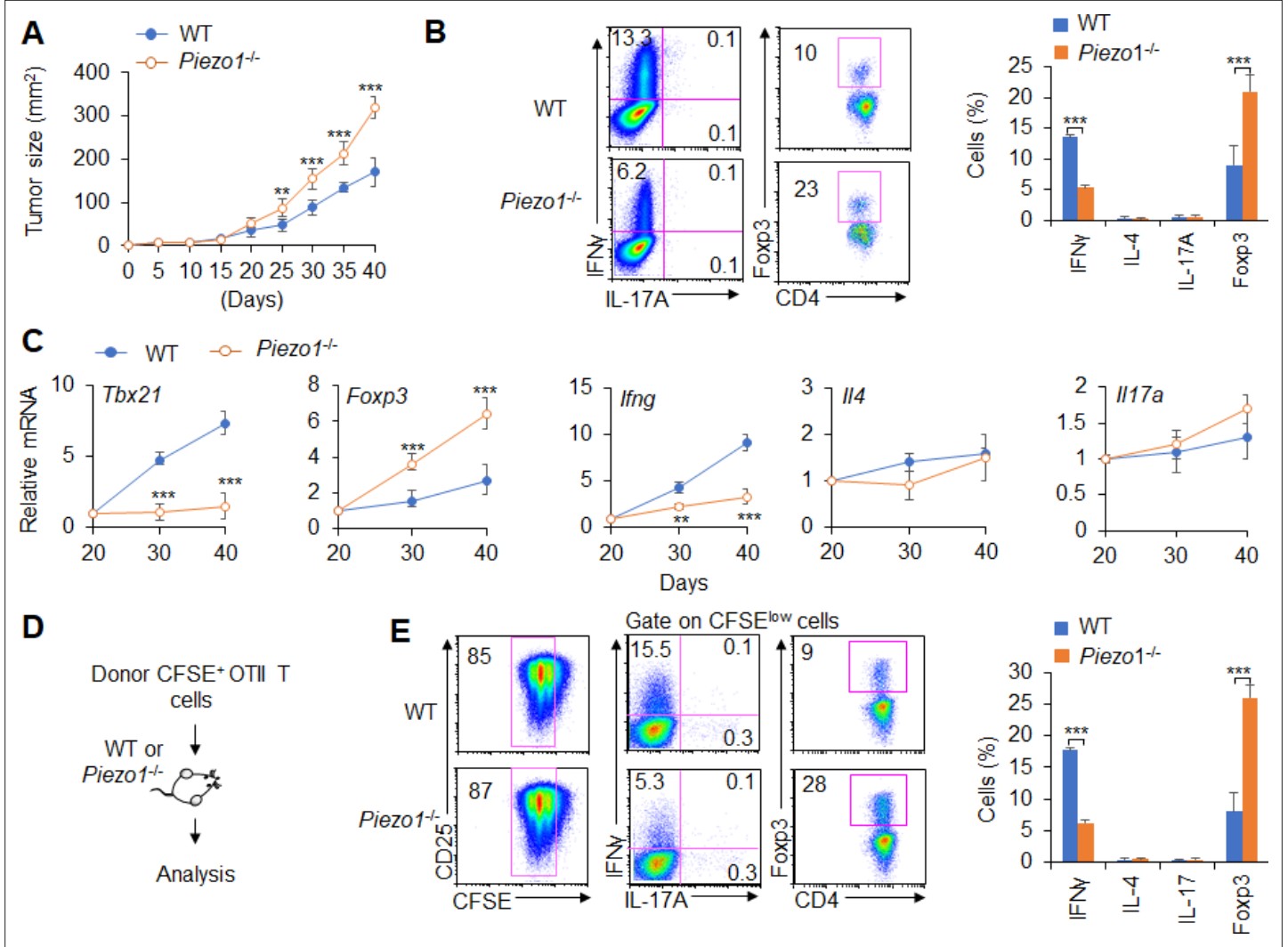

**Figure 1.** Dendritic cell (DC)-specific Piezo1 regulates T cell differentiation in cancer. (**A**) MC38 tumor cells were implanted subcutaneously in WT and *Piezo1⁻/⁻* mice (n=10) and tumor size was measured every 5 days for 40 days. (**B**) Intracellular staining of IFNγ, IL-4, IL-17A, and Foxp3 expression by CD4⁺ T cells sorting from the tumor of WT and *Piezo1⁻/⁻* tumor-bearing mice at day 40. (**C**) mRNA expression of the indicated genes by CD4⁺ T cells isolated from the draining lymph nodes (dLNs) of tumor from WT and *Piezo1⁻/⁻* tumor-bearing mice on the indicated days (the levels in WT mice at day 20 were set to 1). (**D**) MC38 OVA tumor cells were implanted subcutaneously in WT and *Piezo1⁻/⁻* mice (n=10) and at day 20, the CD45.1⁺ donor CFSE⁺OTII CD4⁺T cells were transferred into WT and *Piezo1⁻/⁻* tumor-bearing mice for 10 days. The CD45.1⁺ CFSE⁺ donor T cells from tumors were analyzed and the intracellular staining of IFNγ, IL-4, IL-17A, and Foxp3 expression among CFSE^low donor T cells. The data are representative of three to four independent experiments (mean ± s.d.; n=4). \*\*p<0.01 and \*\*\*p<0.001, compared with the indicated groups.

The online version of this article includes the following source data and figure supplement(s) for figure 1:

**Source data 1.** Tumor size in WT and *Piezo1⁻/⁻* tumor-bearing mice.

**Figure supplement 1.** Piezo1 expressions of dendritic cells (DCs) following innate stimuli.

**Figure supplement 2.** Dendritic cell (DC)-specific Piezo1 deficiency reduced T cell activities in aged mice.

**Figure supplement 3.** Dendritic cell (DC)-specific Piezo1 deficiency alters T cell differentiation in the aged mice.

**Figure supplement 4.** Dendritic cell (DC)-specific Piezo1 regulates T cell differentiation in cancer.

**Figure supplement 5.** Dendritic cell (DC)-specific Piezo1 regulates T cell differentiation in cancer.

absence or presence of antigen induced more Foxp3⁺T_reg cell and less IFNγ⁺ T_H1 cell, even with a variety of *Piezo1⁻/⁻* DCs including splenic CD11b⁺ DC or CD8α⁺ DCs, than WT control DCs (***Figure 3—figure supplement 1***). Moreover, *Tbx21* and *Ifng* expression was significantly downregulated and *Foxp3* expression was significantly upregulated in T cells cocultured with *Piezo1⁻/⁻* DCs. However, *Il4*,

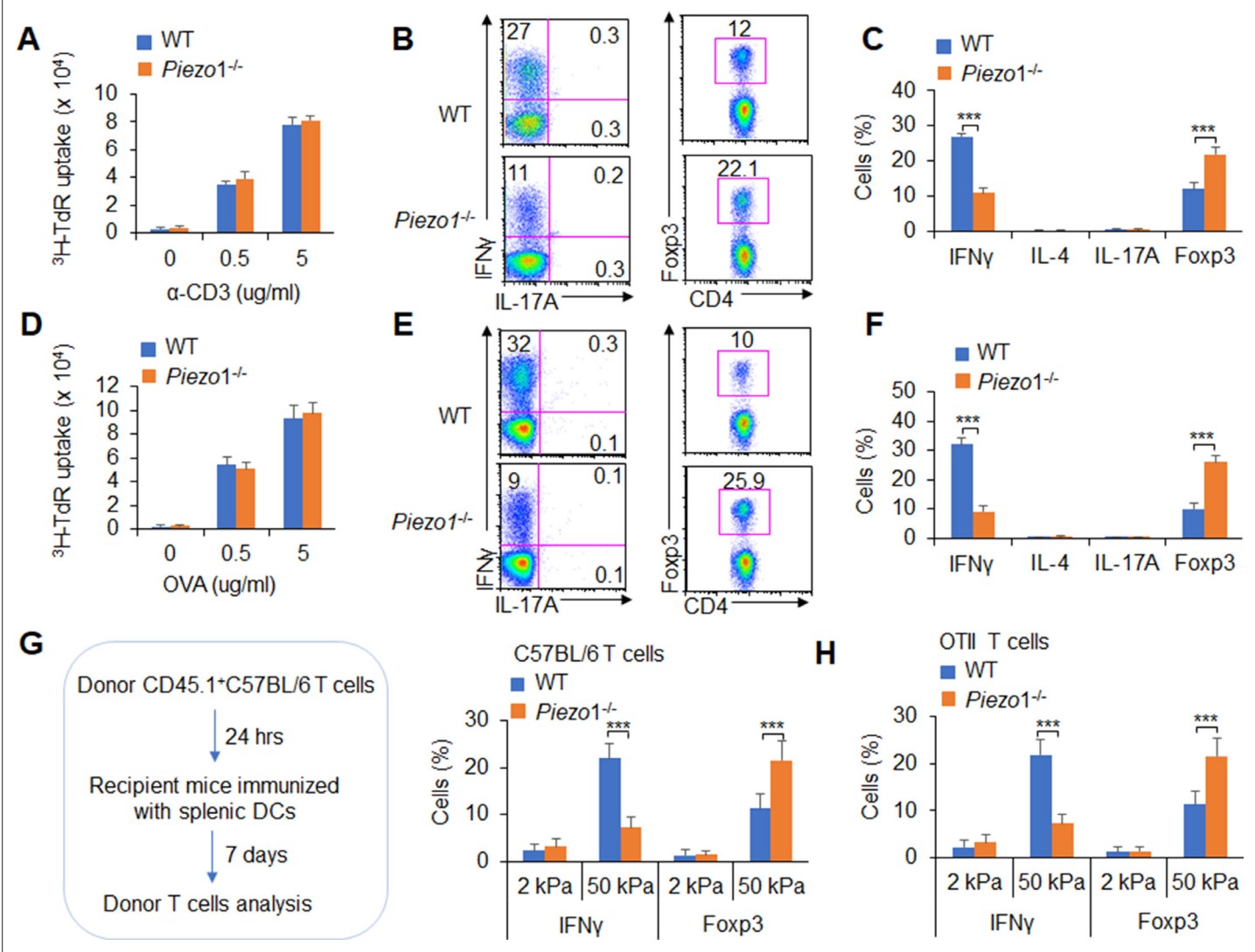

**Figure 2.** Dendritic cell (DC)-specific Piezo1 expression directs the differentiation of $T_H1$ and $T_{reg}$ cells in vivo. (**A–C**) Naïve CD45.2+ T cells were transferred into CD45.1+ C57BL/6 WT mice, and the mice were immunized with WT and *Piezo1−/−* splenic DCs and lipopolysaccharide (LPS). DLN cells were analyzed at day 7 after immunization. (**A**) Donor CD45.2+ T cell proliferation after stimulation with anti-CD3 (2 μg/ml) and anti-CD28 (2 μg/ml) antibodies. (**B–C**) Intracellular staining of IFNγ, IL-4, IL-17A, and Foxp3 expression in donor-derived (CD45.2+) CD4+ T cells after PMA and ionomycin stimulation. A representative figure shown in B, and the data summarized in C. (**D–F**) Naïve OTII T cells were transferred into CD45.1+ C57BL/6 WT mice, and the mice were immunized with WT and *Piezo1−/−* splenic DCs and OVA+CFA. DLN cells were analyzed at day 7 after immunization. (**D**) Donor CD45.2+ T cell proliferation after stimulation with OVA. (**E–F**) Intracellular staining of IFNγ, IL-4, IL-17A, and Foxp3 in donor-derived (CD45.2+) CD4+ T cells after OVA stimulation. A representative figure shown in E, and the data summarized in F. (**G–H**) Splenic DCs isolated from WT and *Piezo1−/−* mice were plated on 2 and 50 kPa hydrogels and incubated for 24 hr. (**G**) Naïve T cells were transferred into CD45.1+ C57BL/6 WT mice, and the mice were immunized with 2 and 50 kPa hydrogel-conditioned DCs. DLN cells were analyzed at day 7 after immunization. Intracellular staining of IFNγ, IL-4, and IL-17A in donor-derived (CD45.2+) CD4+ T cells after PMA and ionomycin stimulationand data summarized. (**H**) Naïve OTII T cells were transferred into CD45.1+ C57BL/6 WT mice, and the mice were immunized with 2 or 50 kPa hydrogel-conditioned DCs and OVA+CFA. DLN cells were analyzed at day 7 after immunization. Intracellular staining of IFNγ, IL-4, and IL-17A in donor-derived (CD45.2+) CD4+ T cells after OVA stimulation and data summarized. The data are representative of three to four independent experiments (mean ± s.d.; n=3–4). ***p<0.001, compared with the indicated groups.

*Il10*, *Il17a* and *Gata3*, *Rorgt* expressions were similar in T cells cocultured with *Piezo1−/−* DCs or WT DCs (**Figure 3—figure supplement 2A**). These data suggest that Piezo1 deficiency in different kinds of DCs including CD11b+ DC and CD8α+ DC subsets regulates the reciprocal differentiation of $T_H1$ and $T_{reg}$ cells in an antigen-specific manner. Coculture of antigen-specific OTII T cells with *Piezo1^{ΔDC}* splenic CD11b+ DCs conditioned by 2 or 50 kPa hydrogels in the presence of antigen induced a higher expression of Foxp3 and a lower expression of IFNγ (**Figure 3—figure supplement 2B-C**).

Together, Piezo1 signals are required for reciprocal $T_H1$ and $T_{reg}$ cell differentiation directed by DCs in an antigen-specific manner.

## DC Piezo1 regulates T cell differentiation through IL-12 and TGFβ1

APCs regulate T cell differentiation by changing costimulatory molecule expression, TCR signaling, and polarizing cytokine production (*Steinman et al., 2003*). Piezo1 deficient does not alter the cell homeostasis level (*Figure 3—figure supplement 3A*), the expression of MHC, the costimulatory molecules CD80, CD86, CD54, PDL1, PDL2, and CCR7 expressions in DCs (*Figure 3—figure supplement 3B-C*). And, the phagocytosis activities of DC to IgG-coated beads is also comparable between WT and *Piezo1⁻ᐟ⁻* DCs (*Figure 3—figure supplement 3D*). Next, we also detected changes in cytokines secreted by splenic DCs, especially polarizing cytokines important for inducing $T_H1$ and $T_{reg}$ cell differentiation. First, we isolated the DC from MLNs at 40 weeks of age mice and found that *Piezo1⁻ᐟ⁻* DCs have less IL-12 (p70) and more TGFβ1 expression (*Figure 3—figure supplement 4A*). These suggest the intestinal environment receives continuous antigen stimulation from microorganisms or food, which may lead to changes in intestinal tension caused by intestinal movements such as food digestion and absorption or inflammatory stimuli, and target the Piezo1 signal of DCs to trigger the differentiation of different subsets of T cells in the aged mice. Next, we found that *Piezo1⁻ᐟ⁻* caused DCs significantly higher TGFβ1 production and lower IL-12 (p40), IL-12 (p70), but not IL-23 production in the presence of LPS (*Figure 3A*, *Figure 3—figure supplement 4B-C*) or 50 kPa, but not 2 kPa, hydrogels (*Figure 3B*). These results together suggest that the polarizing cytokines TGFβ1 and IL-12 are probably involved in regulating T cell subset differentiation induced by Piezo1 in DCs stimulating by inflammatory stimuli or stiffness signals.

We selected a splenic DC-T coculture system to determine whether DC-specific Piezo1 expression regulates T cell subset differentiation through the polarizing cytokines TGFβ1 and IL-12. Although Piezo1 deficiency in DCs caused a significantly lower IFNγ⁺ $T_H1$ cell percentage, adding IL-12 to the coculture system almost completely recovered the proportion of *Piezo1⁻ᐟ⁻* DCs during $T_H1$ cell differentiation (*Figure 3C*). Even DCs conditioned by 50 kPa hydrogel treatment caused similar effects (*Figure 3D*). Consistently, blocking TGFβ1 signaling with an anti-TGFβ1 antibody in the coculture system almost completely recovered the effect of *Piezo1⁻ᐟ⁻* splenic DCs on $T_{reg}$ cell differentiation. Similar effects could be observed when DCs were treated with LPS or 50 kPa hydrogels (*Figure 3C–D*). Thus, we could conclude that DC-specific Piezo1 expression regulates the reciprocal differentiation of $T_H1$ and $T_{reg}$ cells through the polarizing cytokines IL-12 and TGFβ1.

## DC-specific Piezo1 expression induces T cell differentiation through TGFβR2 and IL-12Rβ2

T cell differentiation-inducing cytokines often change the corresponding receptors on the surface of T cells and program the differentiation of T cell subsets (*Zhu et al., 2010*; *Liu et al., 2010*). We further determined the corresponding receptor expression on T cells. Piezo1 deficiency in splenic DCs significantly increased the expression of TGFβR2 and decreased the expression of IL-12Rβ2 but did not affect the expression of TGFβR1/3 or IL-12Rβ1 on T cells in a DC-T coculture system (*Figure 4A*, *Figure 4—figure supplement 1*). Interestingly, Piezo1-deficient DCs also exhibited significantly higher levels of phosphorylated Smad3 and lower levels of phosphorylated STAT4 than WT DCs (*Figure 4B*). These data suggest that TGFβR2-p-Smad3 or IL-12Rβ2-p-STAT4 signaling is involved in the $T_{reg}$ and $T_H1$ cell differentiation induced by DC-specific Piezo1 expression.

To determine whether TGFβR2-pSmda3 and IL-12Rβ2-pSTAT4 are required for promoting the T cell differentiation induced by DC-specific Piezo1 expression, we knocked down IL-12Rβ2 and TGFβR2 expression in T cells with shRNA in a DC-T coculture system (*Figure 4—figure supplement 2*). Although DC-specific Piezo1 deficiency resulted in significantly more Foxp3⁺ $T_{reg}$ cells, higher phosphorylation of Smad3, and less IFNγ⁺ $T_H1$ cells, lower phosphorylation of STAT4, knockdown of TGFβR2 or IL-12Rβ2 expression significantly recovered these effects compared with the WT conditions (*Figure 4C–D*). These data suggest that TGFβR2-pSmda3 and IL-12Rβ2-pSTAT4 signaling in T cells are required for the T cell differentiation induced by DC Piezo1.

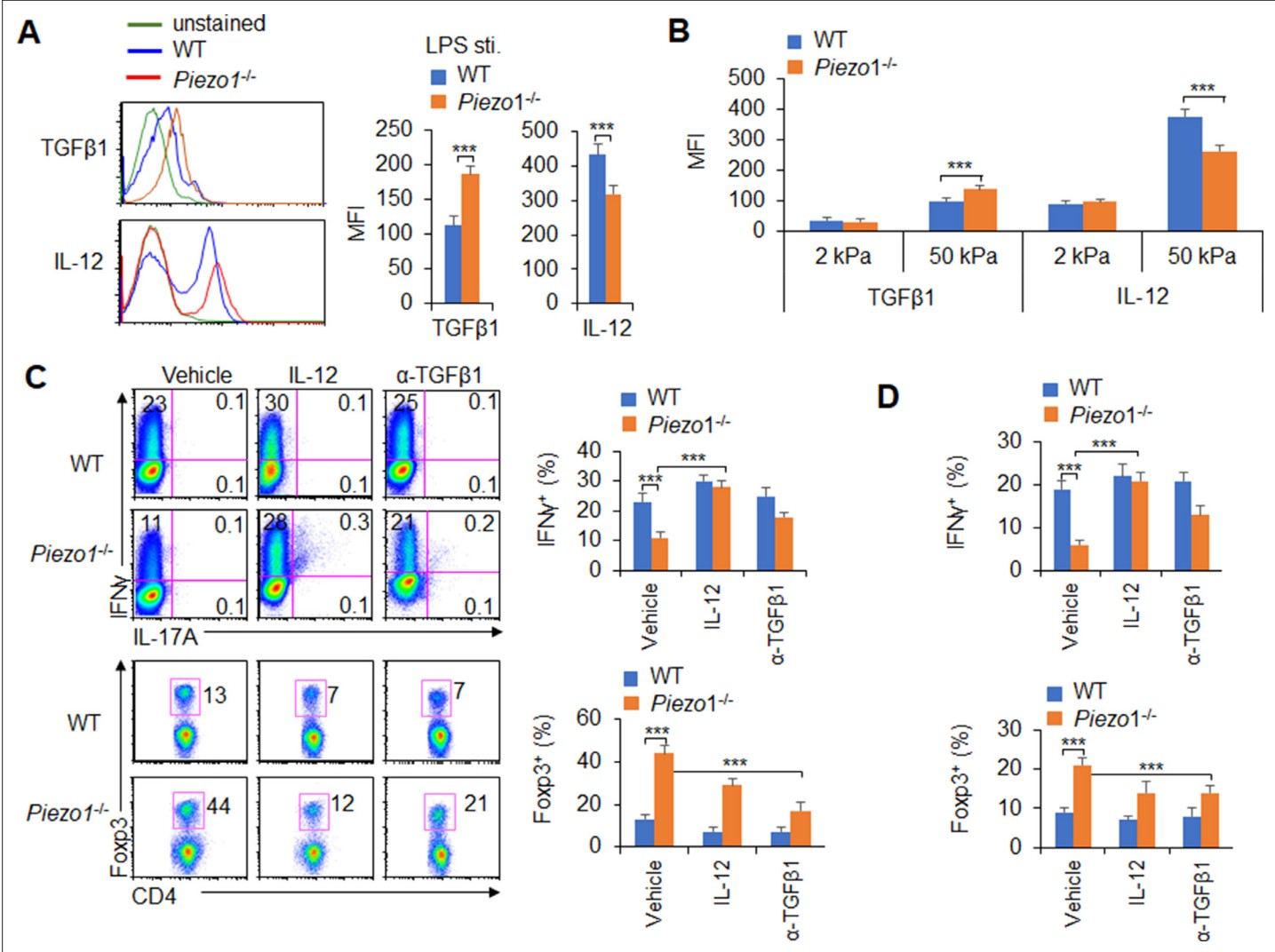

**Figure 3.** Piezo1 regulates IL-12 and TGFβ1 production by dendritic cells (DCs) to direct T_H1 and T_reg cell differentiation. (**A–B**) Intracellular staining of IL-12p40 and TGFβ1 expression in WT and *Piezo1^-/-* splenic DCs after 5 hr of treatment with lipopolysaccharide (LPS) (A; 10 ng/ml) or culture on 2 and 50 kPa hydrogels (**B**). A representative figure shown on the left, and the data summarized on the right. (**C**) Intracellular staining of IFNγ and Foxp3 in T cells cocultured with WT and *Piezo1^-/-* splenic DCs in the presence of the indicated treatments (IL-12, Peprotech, 10 μg/ml or anti-TGFβ1, R&D Systems, 20 μg/ml) for 5 days. A representative figure shown on the left, and the data summarized on the right. (**D**) Intracellular staining of IFNγ (upper panel) and Foxp3 (lower panel) in T cells cocultured with WT and *Piezo1^ΔDC* splenic DCs conditioned with 50 kPa hydrogel and the indicated treatments for 5 days and data summarized. The data are representative of three independent experiments (mean ± s.d.; n=3–5). ***p<0.001, compared with the indicated groups.

The online version of this article includes the following figure supplement(s) for figure 3:

**Figure supplement 1.** Dendritic cell (DC)-specific Piezo1 expression directs T_H1 and T_reg differentiation in vitro.

**Figure supplement 2.** Dendritic cell (DC)-specific Piezo1 expression directs T_H1 and T_reg differentiation in vitro.

**Figure supplement 3.** Piezo1 regulates dendritic cell (DC) homeostasis and function.

**Figure supplement 4.** Piezo1 regulates dendritic cell (DC) cytokine production.

## Piezo1 regulates IL-12 and TGFβ1 production through the SIRT1-HIF1α-glycolysis pathway

How does Piezo1 regulate IL-12 and TGFβ1 production to direct T cell differentiation? To study the mechanisms underlying the effects of Piezo1, splenic DCs were stimulated by LPS, and we assessed the signaling downstream of LPS stimulation, including Erk, c-jun-NH2-kinase (JNK), p38MAPK, SIRT1, HIF1α, and glycolytic molecular signaling.

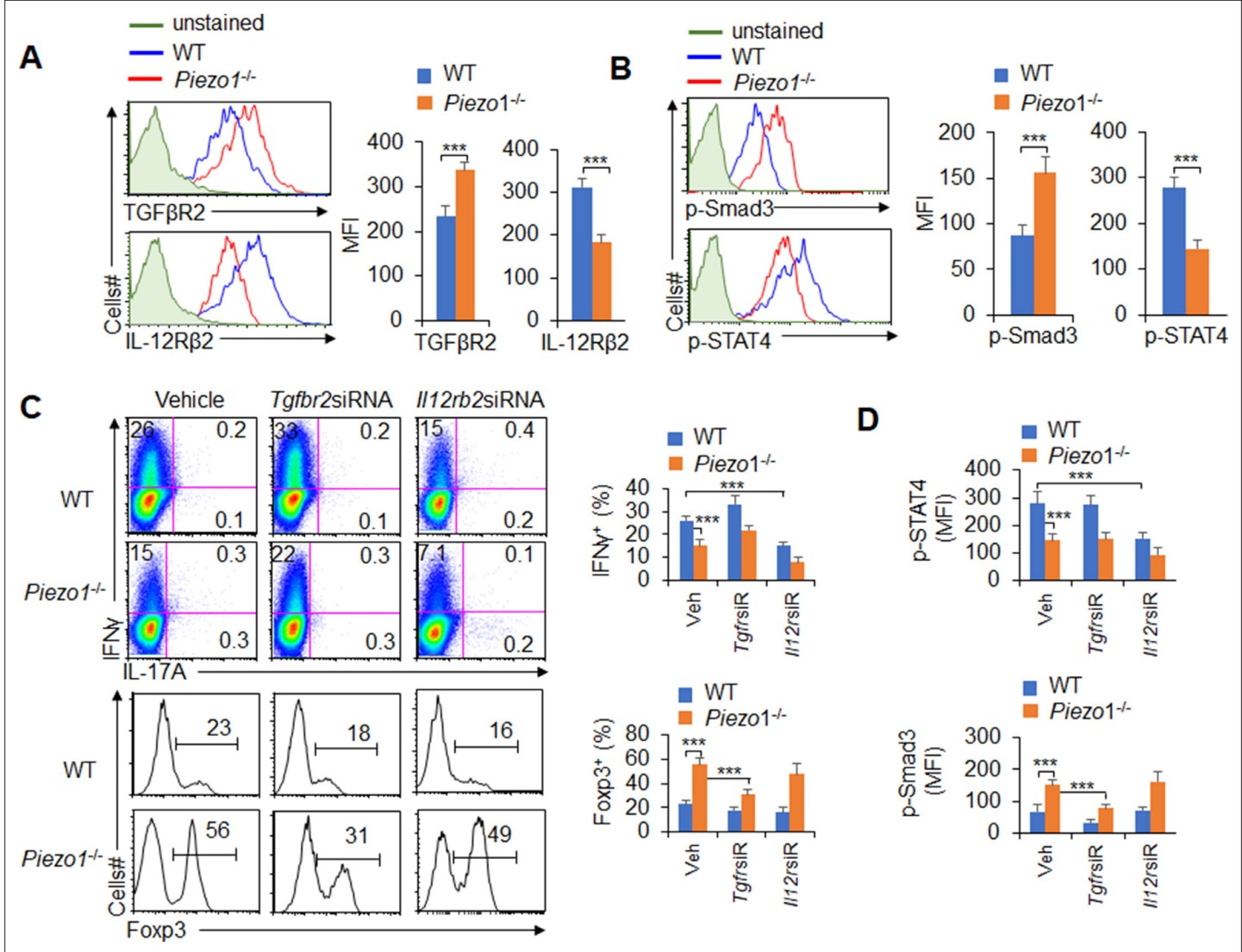

**Figure 4.** TGFβR2-pSmad3 and IL-12Rβ2-pSTAT4 are required for the T cell differentiation induced by dendritic cell (DC)-specific Piezo1 expression. (**A**) Expression of TGFβR2 and IL-12Rβ2 in T cells cocultured with WT or *Piezo1*<sup>-/-</sup> splenic DCs for 5 days. A representative figure shown on the left, and the data summarized on the right. (**B**) Intracellular staining of p-Smad3 and p-STAT4 in T cells cocultured with WT or *Piezo1*<sup>-/-</sup> splenic DCs for 5 days. A representative figure shown on the left, and the data summarized on the right. (**C–D**) Sorted naïve T cells were transfected with control, *Tgfbr2* short hairpin RNA (shRNA) vector, or *Il12rb2* shRNA vector and cocultured with WT or *Piezo1*<sup>-/-</sup> DCs for 5 days. Intracellular staining of IFNγ (C; upper panel) and Foxp3 (C; lower panel) in T cells. A representative figure shown on the left, and the data summarized on the right. (**D**) Intracellular staining of p-Smad3 and p-STAT4 in T cells and data summarized. The data are representative of three independent experiments (mean ± s.d.; n=4). ***p<0.001, compared with the indicated groups.

The online version of this article includes the following figure supplement(s) for figure 4:

**Figure supplement 1.** Expressions of TGFβR1/2 and IL-12Rβ1 in T cells induced by *Piezo1*<sup>-/-</sup> dendritic cells (DCs).

**Figure supplement 2.** Knockdown of *Tgfbr2* short hairpin RNA (shRNA) and *Il12rb2* shRNA in T cells.

We investigated the role of glycolysis and oxidative phosphorylation (OXPHOS) signal activities in the functional regulation of DCs induced by Piezo1. LPS treatment led to an increase in the proton production rate (PPR), but splenic DCs treated with the Piezo1 agonist Yoda1 exhibited significantly enhanced PPR values and expression of glycolytic molecules but not oxygen consumption rates (OCRs) (*Figure 5—figure supplement 1*). Blocking glycolysis with 2-deoxy-D-glucose (2-DG), a prototypical inhibitor of glycolysis pathways, significantly recovered the IL-12 and TGFβ1 production in splenic DCs induced by Yoda1 treatment (*Figure 5—figure supplement 2*). Furthermore, Piezo1 deficiency in DCs significantly decreased the PPR value and the expression of glycolytic molecules but not the OCR

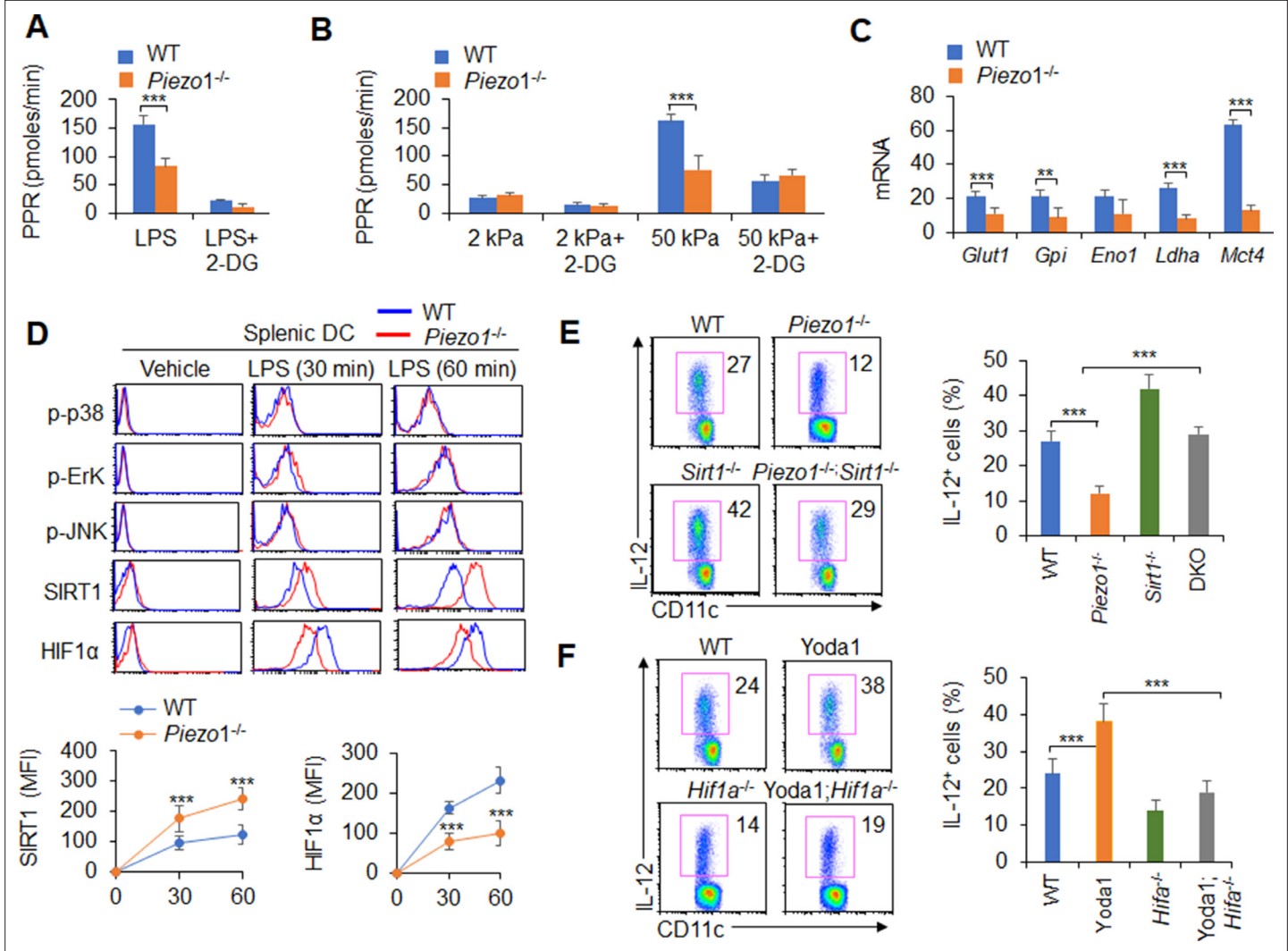

**Figure 5.** Piezo1 regulates TGFβ1 and IL-12 production through the SIRT1-HIF1α-glycolysis pathway. (**A–B**) Sorted splenic dendritic cells (DCs) from WT or *Piezo1*-/- mice were stimulated with lipopolysaccharide (LPS) (10 ng/ml; **A**) or with 2 or 50 kPa hydrogels (**B**) for 24 hr in the presence or absence of 2-deoxy-D-glucose (2-DG) (1 mmol/l). The proton production rate (PPR) was analyzed as a readout for glycolysis. (**C**) mRNA expression of glycolytic molecules in splenic DCs from WT or *Piezo1*-/- mice treated with LPS (10 ng/ml) for 12 hr. The levels in the WT control group were set to 1. (**D**) Intracellular staining of p38, Erk, and JNK phosphorylation and SIRT1 and HIF1α expression in splenic DCs from WT or *Piezo1*-/- mice. A representative figure shown in the upper panel, and the data summarized in the lower panel. (**E**) Splenic DCs from WT, *Piezo1*-/- , *Sirt1*-/-, and *Piezo1/Sirt1* double knockout (DKO; *Piezo1*-/-*Sirt1*-/-) mice were stimulated with LPS (10 ng/ml). Intracellular staining of IL-12p40. A representative figure shown on the left, and the data summarized on the right. (**F**) Splenic DCs from WT or *Hif1a*-/- mice were stimulated with LPS (10 ng/ml) in the presence or absence of Yoda1 (25 μM). Intracellular staining of IL-12p40. A representative figure shown on the left, and the data summarized on the right. The data are representative of three independent experiments (mean ± s.d.; n=3–4). **p<0.01 and ***p<0.001, compared with the indicated groups.

The online version of this article includes the following figure supplement(s) for figure 5:

**Figure supplement 1.** Piezo1 alters glycolytic metabolic signaling activities of dendritic cells (DCs).

**Figure supplement 2.** Glycolysis activities are required for IL-12 and TGFβ1 production in dendritic cells (DCs) induced by Yoda1 treatment.

**Figure supplement 3.** Glycolytic metabolism activities are required for IL-12 and TGFβ1 production in dendritic cells (DCs) induced by Piezo1 deficiency.

**Figure supplement 4.** TGFβ1, HIF1α, and glycolysis activity alteration in dendritic cells (DCs) induced by Piezo1 and SIRT1 deficiency.

**Figure supplement 5.** TGFβ1, SIRT1, and glycolysis activity alteration in dendritic cells (DCs) induced by Piezo1-HIF1α signaling.

value (*Figure 5A–C*, *Figure 5—figure supplement 3A*). Blocking glycolysis with 2-DG significantly recovered the productions of IL-12 and TGFβ1 in *Piezo1⁻/⁻* DCs to normal level compared with WT DCs (*Figure 5A–C*, *Figure 5—figure supplement 3B-C*). These data altogether suggest that glycolysis activities are required for the polarizing cytokine production in DCs induced by Piezo1.

Additionally, as expected, LPS activated all downstream pathways in WT DCs including p-JNK, p-Erk, p-p38, SIRT1, and HIF1α signal pathway. However, Piezo1 deletion in DCs enhanced the phosphorylation of Erk, p38, and JNK, similar to WT DCs. However, *Piezo1⁻/⁻* DCs exhibited stronger and sustained activation of the histone deacetylase SIRT1 and weaker and shorter activation of the transcription factor HIF1α (*Figure 5D*). Thus, Piezo1 is probably associated with SIRT1-HIF1α signaling and glycolysis activation.

To determine whether SIRT1 is involved in this regulation, we used Piezo1 and SIRT1 double knockout (DKO) mice in this investigation. Interestingly, less IL-12 and more TGFβ1 production in *Piezo1⁻/⁻* DCs was significantly reversed in Piezo1-SIRT1 DKO cells (*Figure 5E*, *Figure 5—figure supplement 4A*). Consistently, HIF1α expression and glycolysis activities were significantly recovered to a normal level (*Figure 5—figure supplement 4B-C*). These data suggest that SIRT1 is required for the IL-12 and TGFβ1 production in DCs induced by Piezo1 and HIF1α and glycolysis activation is probably related with these alterations.

To determine whether HIF1α is involved in this regulation, we crossed DC HIF1α conditional knockout mice (*Hif1a⁻/⁻*) with *Hif1a*^flox/flox and *Cd11c-Cre* mice. Splenic DCs were isolated from WT and *Hif1a⁻/⁻* mice and treated with the Piezo1 agonist Yoda1. More IL-12 and less TGFβ1 in Yoda1-treated DCs was significantly reversed in Yoda1-treated *Hif1a⁻/⁻* DCs (*Figure 5F*, *Figure 5—figure supplement 5A*). These data suggest that HIF1α is required for the IL-12 and TGFβ1 production in DCs induced by Piezo1.

Moreover, splenic DCs were treated with the Piezo1 agonist Yoda1, which significantly altered glycolysis activity and SIRT1 expression in DCs. SIRT1 expressions cannot be altered in *Hif1a⁻/⁻* DCs (*Figure 5—figure supplement 5B*). However, treatment of *Hif1a⁻/⁻* with Yoda1 significantly reversed the alteration of glycolysis but not SIRT1 expression (*Figure 5—figure supplement 5B-C*). Thus, HIF1α and glycolysis is downstream targets of SIRT1 in regulating the Piezo1-induced cytokine production in DCs.

## Piezo1 regulates IL-12 and TGFβ1 production through the calcium-calcineurin-NFAT axis

Mechanically activated ion channel Piezo1 regulates macrophage or DC function by altering calcium permeability (*Atcha et al., 2021*; *Chakraborty et al., 2021*). Therefore, we assessed the level of calcium influx in *Piezo1⁻/⁻* splenic DCs. Inflammatory or stiffness stimulation caused a significant decrease in the calcium influx in *Piezo1⁻/⁻* DCs compared with the WT control (*Figure 6A*). And, Piezo1 agonist Yoda1 treatment significantly enhanced intracellular calcium influx, decreased TGFβ1 secretion, and increased IL-12 secretion by DCs exposed to LPS or conditioned by 50 kPa hydrogels (*Figure 6B–D*). Importantly, blocking the Ca²⁺ signaling with ruthenium red reversed these alterations. But, another nonspecific ion channel inhibitor, gadolinium, had no effects on calcium influx or cytokine production in DCs (*Figure 6B–D*). Consistently, *Piezo1⁻/⁻* DCs showed lower calcium influx and exhibited higher TGFβ1 and lower IL-12 levels, and blocking the Ca²⁺ signaling pathway with ruthenium red, but not gadolinium reversed these alterations in cytokine production (*Figure 6—figure supplement 1*). Thus, calcium signaling pathway is required for the TGFβ1 and IL-12 production in DCs induced by Piezo1.

Previous studies have shown that calcineurin-NFAT are critical molecules of the calcium signaling pathway in regulating the immune response (*Wang et al., 2015*; *Rudensky et al., 2006*; *Hu et al., 2007*; *Vaeth et al., 2012*). These prompted us to investigate whether calcineurin-NFAT signals are necessary for Piezo1 to regulate DC function through calcium signaling pathway. Therefore, we pharmacologically targeted calcineurin and NFAT to assess their roles in regulating Piezo1-induced TGFβ1 and IL-12 production in DCs. Splenic DCs treated with the Piezo1 agonist Yoda1 caused more IL-12 and less TGFβ1 production, but blocking calcineurin with its inhibitor cyclosporin A (CsA) reversed these alterations in DCs (*Figure 6E*, *Figure 6—figure supplement 2A*). These data suggest that calcium-calcineurin signaling are probably required for cytokine productions in DCs induced by Piezo1.

As reported (*Wang et al., 2015*; *Rudensky et al., 2006*; *Hu et al., 2007*; *Vaeth et al., 2012*), NFAT is critical transcriptional factor for regulating calcium-calcineurin signaling pathway in mediating

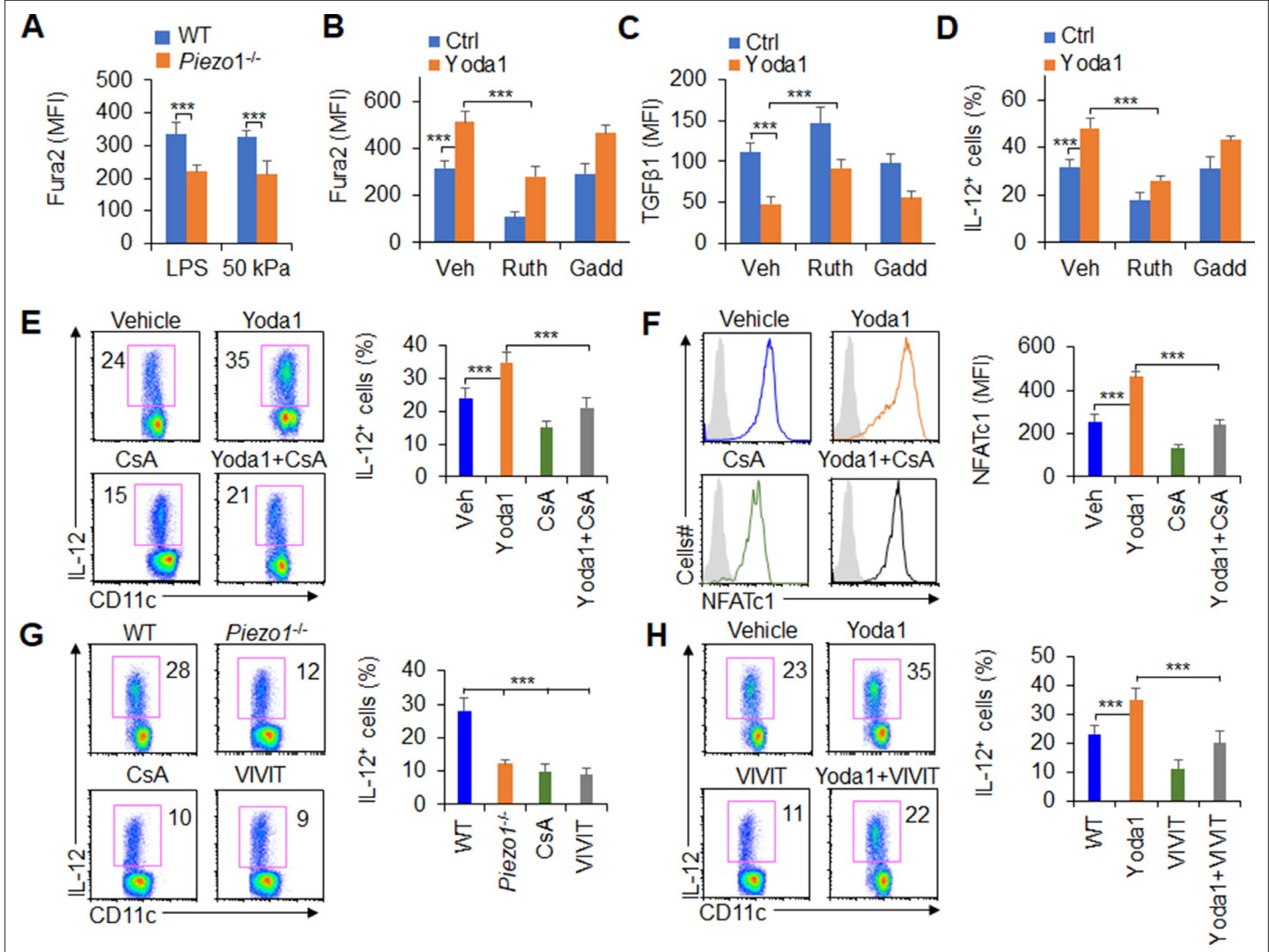

**Figure 6.** Piezo1 regulates TGFβ1 and IL-12 production through the calcium-calcineurin-NFAT axis. (**A**) Measurement of intracellular $Ca^{2+}$ concentrations with Fura2 dye in splenic dendritic cells (DCs) from WT or *Piezo1*[-/-] mice treated with lipopolysaccharide (LPS) (10 ng/ml) or cultured on plates containing 50 kPa hydrogels. (**B**) Intracellular $Ca^{2+}$ concentrations measured with Fura2 in splenic DCs from WT mice after the indicated treatment (Yoda1, 25 µM, MCE; ruthenium red, 30 µM, Sigma; gadolinium chloride, 10 µM, Sigma). (**C–D**) Intracellular staining of TGFβ1 (**C**) and IL-12p40 (**D**) in splenic DCs from WT mice after the indicated treatments. (**E**) Intracellular staining of IL-12p40 in splenic DCs from WT mice after the indicated treatments. A representative figure shown on the left, and the data summarized on the right. (**F**) Intracellular staining of NFATc1 in splenic DCs from WT mice after the indicated treatments (CsA, 10 nM). A representative figure shown on the left, and data summarized on the right. (**G–H**) Intracellular staining of IL-12p40 in splenic DCs from WT or *Piezo1*[-/-] mice after the indicated treatments (Yoda1, 25 µM, MCE; 11R-VIVIT, 100 nM, MCE; CsA, 10 nM, Sigma). A representative figure shown on the left, and data summarized on the right. The data are representative of three to four independent experiments (mean ± s.d.; n=3–4). ***p<0.001, compared with the indicated groups.

The online version of this article includes the following figure supplement(s) for figure 6:

**Figure supplement 1.** Piezo1 regulates TGFβ1 and IL-12 production through calcium signaling.

**Figure supplement 2.** Piezo1 regulates TGFβ1 production through calcium-calcineurin-NFAT axis.

immune cell activities. Moreover, upregulation of Piezo1 with Yoda1 treatment enhanced the expression of NFAT, blocking calcineurin with CsA inhibits the expression of NFAT (*Figure 6F*). To test the role of NFAT for Piezo1 to regulate DC function, we pharmacologically targeted NFAT with its inhibitor VIVT to assess the role of NFAT in regulating Piezo1-induced TGFβ1 and IL-12 production by DCs. *Piezo1*[-/-] DCs exhibited less IL-12 and more TGFβ1, and blocking calcineurin with CsA and blocking NFAT with VIVIT consistently showed similar cytokine production by DCs (*Figure 6G*, *Figure 6—figure supplement 2B*). Interestingly, splenic DCs treated with the Piezo1 agonist Yoda1 exhibited

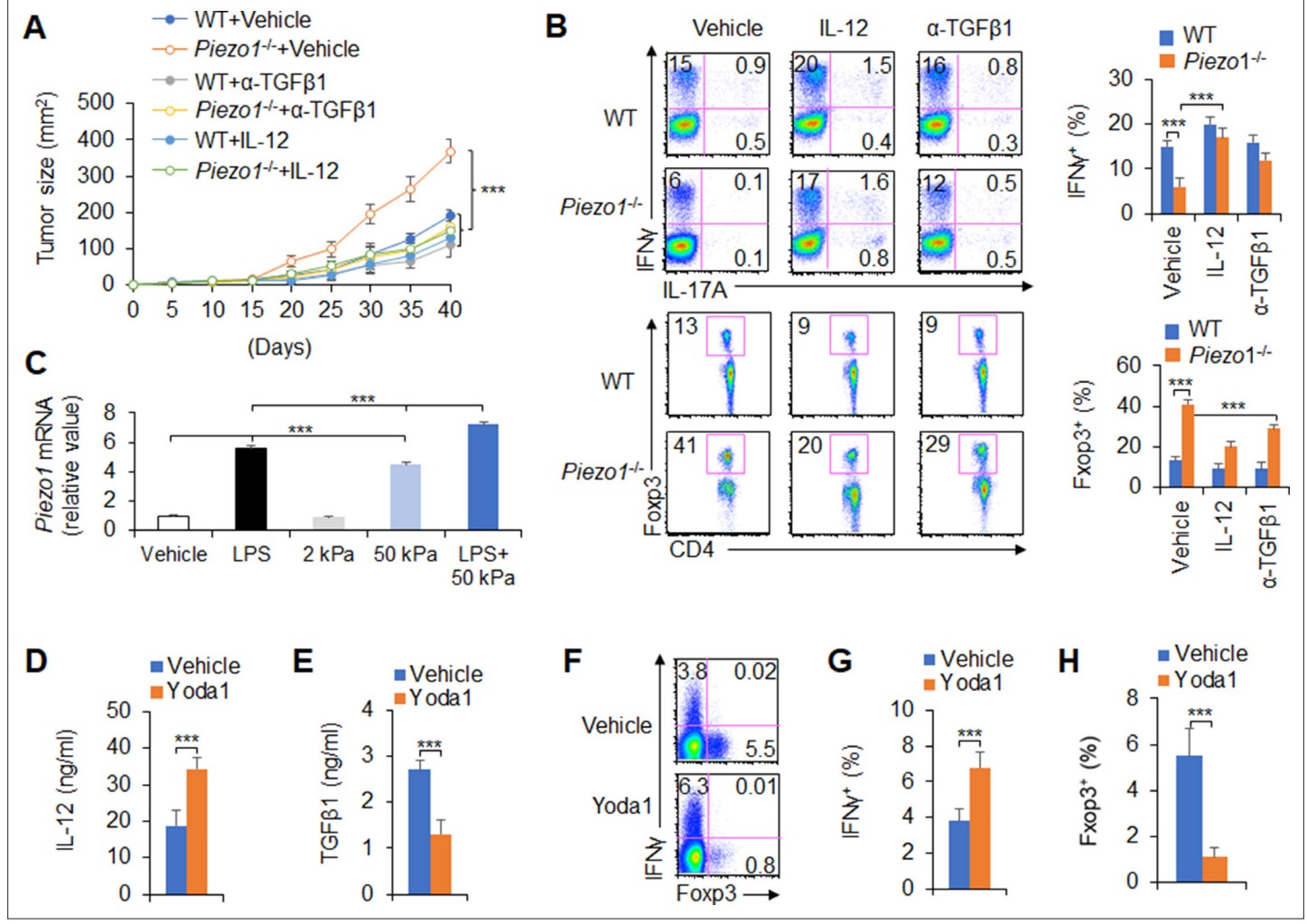

**Figure 7.** IL-12 and TGFβ1 are critical for dendritic cell (DC) piezo1-dependent T cell differentiation in promoting cancer growth. (**A**) MC38 tumor cells were implanted subcutaneously in WT and *Piezo1⁻/⁻* mice (n=10), IL-12 100 ng or anti-TGFβ1 mAb 200 ng per mouse in 50 μl volume or vehicle (PBS) was locally injected into tumor once a week and tumor size was measured every 5 days for 40 days. (**B**) Intracellular staining of IFNγ, IL-17A, and Foxp3 expression in CD4⁺ T cells from the tumor of WT and *Piezo1⁻/⁻* tumor-bearing mice at day 40. (**C**) *Piezo1* mRNA expression of human DCs with the indicated treatment (lipopolysaccharide [LPS], 10 ng/ml or conditioned with 2 or 50 kPa hydrogels plate or LPS+50 kPa hydrogels plate). Levels in the vehicle group were set to 1. (**D–E**) IL-12p70 (**D**) and TGFβ1 (**E**) production of human DCs treated by LPS (10 ng/ml) for 5 hr. (**F–H**) Human DCs pulsed with LPS (10 ng/ml) were cocultured with human T cells for 5 days, in the absence or presence of Yoda1 (25 μM). The intracellular staining of IFNγ and Foxp3 in T cells. ***p<0.001 compared with the indicated groups. Data are representative of three independent experiments (mean ± s.d.; n=3–4). ***p<0.001, compared with the indicated groups.

The online version of this article includes the following figure supplement(s) for figure 7:

**Figure supplement 1.** Dendritic cell (DC) Piezo1 controls the differentiation of T$_H$1 and T$_{reg}$ cells in cancer.

---

more IL-12 and less TGFβ1 production, but blocking NFAT with VIVIT reversed these alterations in DCs (*Figure 6H*, *Figure 6—figure supplement 2C*). Altogether, these data suggest that the calcium-calcineurin-NFAT axis is required for regulating IL-12 and TGFβ1 production by DC Piezo1.

## IL-12 and TGFβ1 are critical for DC Piezo1-dependent T cell differentiation in promoting cancer growth

To test the significance of DC Piezo1-dependent T cell differentiation in anti-tumor immunity, IL-12 and anti-TGFβ1 antibody were locally injected into the tumor once a week to treat tumor-bearing mice. Although the rate of tumor growth was significantly faster and greater in *Piezo1⁻/⁻* than in WT mice, IL-12 or anti-TGFβ1 antibody treatment significantly inhibited the tumor growth caused

by *Piezo1*[-/-] (*Figure 7A*). Consistently, *Piezo1*[-/-] tumor-bearing mice had more Foxp3[+] T$_{reg}$ cells and fewer IFNγ[+] T$_H$1 cells in tumor tissue compared with WT control. However, IL-12 or anti-TGFβ1 antibody treatment reversed these alterations induced by *Piezo1*[-/-] (*Figure 7B*). Thus, these data suggest that IL-12 and TGFβ1 are critical for DC Piezo1-dependent T cell differentiation in regulating cancer growth.

Next, we test to apply a pharmacological approach to target Piezo1 in human DCs and determine whether we can recapitulate our finding in genetic targeting Piezo1. Piezo1 expressions were determined in human DC cells, which is from human peripheral blood monocytes. Inflammatory LPS or 50 kPa-conditioned hydrogels alone or together significantly upregulated Piezo1 expressions (*Figure 7C*). Further, we applied Piezo1 agonist Yoda1 to human DC-T cell coculture system, whereby T cells were isolated from human cord blood. The pharmacological activation of Piezo1 in human DCs largely recapitulated what we observed in genetic mouse DCs in terms of the production of IL-12 and TGFβ1 in human DC (*Figure 7D–E*) and the alteration of Foxp3[+]T$_{reg}$ and IFNγ[+]T$_H$1 in human T cells (*Figure 7F–H*). Thus, our data demonstrated that Piezo1 mediated an evolutionary conserved signaling pathway in both mouse and human DCs.

## Discussion

DCs play a central role in initiating first-line innate immunity and inducing subsequent adaptive immunity in protecting against tumorigenesis (*Medzhitov et al., 2011*; *Steinman, 2012*). As a professional APCs, DCs can efficiently shape antigen-specific adaptive immune responses by presenting various exogenous and endogenous antigen stimuli, regulating cell surface costimulatory molecule expression, and producing cytokines and chemokines (*Joffre et al., 2009*; *Iwasaki and Medzhitov, 2010*). Innate inflammatory stimuli include infectious factors, oxygen, nutrient availability, and even force and pressure, often change the DC responses and affect the immune outcome in diseases. Especially, tumor microenvironment usually integrates different innate inflammatory and stiffness stimuli and develop a complex stimulation microenvironment, but how does DC respond to inflammatory and stiffness stimuli and regulates T cell differentiation in tumor remains unclear. Here, our data revealed that the mechanical sensor Piezo1, a signal node, responds to innate inflammatory and/or stiffness stimuli and integrates both the SIRT1-HIF1α-glycolysis metabolic signaling axis and calcium-calcineurin-NFAT signaling in DCs to drive T$_H$1 differentiation while inhibiting T$_{reg}$ lineage commitment in inhibiting tumor growth in the context of complex tumor microenvironment. The changes in IL-12Rβ2/TGFβR2 expression and downstream STAT4/SMAD3 signaling in responding T cells further result in strong DC-T cell crosstalk, indicating the differentiation of T$_{reg}$ and T$_H$1 cells (*Figure 7—figure supplement 1*). Thus, our results contribute to a more comprehensive understanding of the immunopathological process of DC Piezo1-directed T cell differentiation in the tumor microenvironment.

Recent studies have suggested that Piezo1 is involved in regulating many of diseases, including the infectious inflammation and cancer (*Nguetse et al., 2020*; *Choi et al., 2019*; *Nonomura et al., 2018*; *Chang et al., 2019*; *Xie et al., 2022*; *Kuriyama et al., 2022*; *O'Callaghan et al., 2022*; *Liu et al., 2021*). Piezo1 modulates macrophage polarization and stiffness sensing, which are related to calcium influx and the promotion of macrophage activation by actin (*Atcha et al., 2021*). The Piezo1-mediated response to LPS inflammatory stimulation regulates cell activation (*Geng et al., 2021*). Global inhibition of Piezo1 with a peptide inhibitor showed protective effects against both cancer and septic shock (*Aykut et al., 2020*). In addition to inflammatory stimulation, Piezo1 also showed immune regulatory effects on mechanical signals. Cyclical hydrostatic pressure initiates an inflammatory response via the mechanically activated ion channel Piezo1 (*Solis et al., 2019*). Additionally, mechanical stiffness controls DC metabolism and function (*Chakraborty et al., 2021*). Although these studies have clearly shown that Piezo1 respond to inflammatory stimulation as well as induce an immune response to mechanical stimulation and regulate immune cell functions, especially in innate immune cells, it is still unclear how to target DCs, to direct T cell differentiation in cancer. Our results showed that Piezo1 responds to inflammatory stimulation or stiffness signals, and subsequently, Piezo1 effectively integrates metabolism signals and ion signals pathway including SIRT1-HIF1α-glycolysis and the calcium-calcineurin-NFAT signaling pathway to direct the differentiation of T$_{reg}$ and T$_H$1 cells by regulating the production of DC-derived polarizing cytokines, including IL-12 and TGFβ1, in the context of tumor microenvironment (*Figure 7—figure supplement 1*).

Transcriptional factor HIF1α has been implicated as a critical proinflammatory signaling module in myeloid leukocytes (*Cramer et al., 2003*; *Rius et al., 2008*; *Sandau et al., 2001*; *Dang et al., 2011*; *Zhao et al., 2009*; *McInturff et al., 2012*). Consistent with recent findings that SIRT1 is responsible for the deacetylation and destabilization of HIF1α (*Lim et al., 2010*; *Laemmle et al., 2012*). HIF1α is critically involved in regulating Piezo1-induced innate immune cell function (*Solis et al., 2019*). The metabolic mechanism is probably critical in regulating DC function (*Chakraborty et al., 2021*). HIF1α-dependent glycolysis metabolism is also critical for regulating $T_H9$ differentiation and MDSC development and function (*Wang et al., 2016*). The present data showed that Piezo1 could target SIRT1-HIF1α-glycolysis metabolism signaling to modulate DC-derived polarizing cytokine secretion. Ion channel Piezo1 is sensitive to calcium influx in regulating immune cell function (*Atcha et al., 2021*). These data also showed that blocking calcium influx significantly altered Piezo1-mediated DC-derived cytokine secretion. Previous studies have displayed the roles of calcineurin-NFAT in MDSCs in regulating $T_{reg}$ function (*Wang et al., 2015*). Here, we further showed that Piezo1 targeting the calcineurin-NFAT axis modulates DC-derived polarizing cytokine production to direct $T_{reg}$ and $T_H1$ cell differentiation in cancer.

Metabolic regulation and cellular signals are closely and generally associated with the immune response, but there are still few studies on the regulation of the calcium signaling pathway and immune response (*Wang and Green, 2012b*; *Gerriets and Rathmell, 2012*; *Pearce et al., 2013*). Our data showed that Piezo1 integrate the metabolic signaling and calcium signaling pathways to modulate DC-derived polarizing cytokine production in the context of cancerous inflammation. Effective immune responses require DCs to function under various conditions, including altered extracellular mechanical tension states, intracellular metabolic states, and ion levels (possibly caused by inflammatory stimulation) or due to migration, nutritional, and/or hypoxic environments (tumor microenvironment). The adaptation of DCs to changing metabolic states and calcium signaling results from a mechanism of a 'mechanical sensor checkpoint', an active signaling process involved in sensing changes in metabolic and intracellular calcium levels and subsequent signaling transduction and execution (*Wang and Green, 2012a*; *Kaelin, 2008*; *Semenza, 2012*; *Imai et al., 2000*). Our data further suggested that the mechanical sensor Piezo1 in DCs requires the interplay of metabolic and intracellular calcium checkpoints, including the metabolic SIRT1-HIF1α-glycolysis pathway and a sensitive calcium signaling pathway, calcium-calcineurin-NFAT signaling. Therefore, Piezo1 modulator provides a new choice in the research of tumor immune microenvironment intervention, which takes DCs as the target to regulate T cell responses. For examples, Yoda1, a Piezo1 activator, can be used to respond to inflammatory and stiffness signals in tumors, and integrate a variety of intracellular signals to effectively direct T cell differentiation in protecting against tumor growth.

# Materials and methods

## Key resources table

| Reagent type (species) or resource | Designation | Source or reference | Identifiers | Additional information |
|---|---|---|---|---|
| Genetic reagent (*Mus musculus*) | Mouse: C57BL/6J (CD45.1 and CD45.2) | Jackson Laboratory | RRID:IMSR_JAX:000664 | |
| Genetic reagent (*Mus musculus*) | Mouse: *Piezo1*$^{flox/flox}$ | Jicuiyaokang company of China | | |
| Genetic reagent (*Mus musculus*) | Mouse: *Cd11c*-Cre | Nanfangmoshi biological company of China | | |
| Genetic reagent (*Mus musculus*) | Mouse: *Piezo1*$^{-/-}$ | Jicuiyaokang company of China | | |
| Genetic reagent (*Mus musculus*) | Mouse: *Sirt1*$^{flox/flox}$ | Jackson Laboratory | | |
| Genetic reagent (*Mus musculus*) | Mouse: *Hif1a*$^{flox/flox}$ | Jackson Laboratory | | |

*Continued on next page*

*Continued*

| Reagent type (species) or resource | Designation | Source or reference | Identifiers | Additional information |
|---|---|---|---|---|
| Genetic reagent (*Mus musculus*) | Mouse: *Cd4*-Cre | Jackson Laboratory | | |
| Genetic reagent (*Mus musculus*) | Mouse: OTII TCR-transgenic mice | Jicuiyaokang company of China | | |
| Biological sample (*Mus musculus*) | Primary mouse splenocyte cells | Beijing Normal University | | Freshly isolated from mice |
| Biological sample (*Mus musculus*) | Primary mouse bone marrow cells | Beijing Normal University | | Freshly isolated from mice |
| Biological sample (*Mus musculus*) | Primary mouse serum | Beijing Normal University | | Freshly isolated from mice |
| Cell line | Mouse colon cancer cell line MC-38 | China cell bank ATCC Center | MC38 | |
| Cell line | Mouse melanoma cell line B16.F10 | China cell bank ATCC Center | B16.F10 | |
| Cell line | Human DC | Lonza | CC-2701 | |
| Cell line | Human cord blood CD4$^+$ T cells | Lonza | 2C-200 | |
| Antibody | Rat monoclonal anti-mouse CD11c FITC | Thermo Fisher Scientific | Cat # MHCD11C01; RRID: AB_10373970 | FCS (1:100) |
| Antibody | Rat monoclonal anti-mouse CD11c APC eFluor780 | Thermo Fisher Scientific | Cat# 47-0116-42 | FCS (1:100) |
| Antibody | Rat monoclonal anti-mouse CD11c PE | Thermo Fisher Scientific | Cat# 12-0114-81 | FCS (1:100) |
| Antibody | Rat monoclonal anti-mouse CD4 APC-Cy7 | BD Bioscience | Cat#130-109-536 | FCS (1:100) |
| Antibody | Rat monoclonal anti-mouse CD8α FITC | Thermo Fisher Scientific | Cat# 11-0081-82 | FCS (1:100) |
| Antibody | Rat monoclonal anti-mouse CD11b FITC | Thermo Fisher Scientific | Cat# 11-0112-82 | FCS (1:100) |
| Antibody | Rat monoclonal anti-mouse Ly6G PE | Thermo Fisher Scientific | Cat# 12-9668-82 | FCS (1:100) |
| Antibody | Rat monoclonal anti-mouse F4/80 PE | Thermo Fisher Scientific | Cat# 12-4801-82 | FCS (1:100) |
| Antibody | Rat monoclonal anti-mouse CD19 PE | Thermo Fisher Scientific | Cat# 12-0199-42 | FCS (1:100) |
| Antibody | Rat monoclonal anti-mouse TCR FITC | Thermo Fisher Scientific | Cat# TCR2730 | FCS (1:100) |
| Antibody | Rat monoclonal anti-mouse CD44 FITC | Thermo Fisher Scientific | Cat# 11-0441-82 | FCS (1:100) |
| Antibody | Rat monoclonal anti-mouse CD62L APC | Thermo Fisher Scientific | Cat# 11-0621-82 | FCS (1:100) |
| Antibody | Rat monoclonal anti-mouse CD80 APC | Thermo Fisher Scientific | Cat# 17-0801-82 | FCS (1:100) |
| Antibody | Rat monoclonal anti-mouse CD54 FITC | Thermo Fisher Scientific | Cat# 17-0549-42 | FCS (1:100) |
| Antibody | Rat monoclonal anti-mouse MHCII APC | Thermo Fisher Scientific | Cat# 17-5320-82 | FCS (1:100) |
| Antibody | Rat monoclonal anti-mouse CD45 APC | Thermo Fisher Scientific | Cat# 17-0459-42 | FCS (1:200) |

*Continued on next page*

*Continued*

| Reagent type (species) or resource | Designation | Source or reference | Identifiers | Additional information |
|---|---|---|---|---|
| Antibody | Rat monoclonal anti-mouse NK1.1 PE | Thermo Fisher Scientific | Cat# 25-5941-82 | FCS (1:200) |
| Antibody | Rat monoclonal Anti-mouse CCR7 APC | Thermo Fisher Scientific | Cat# A18389 | FCS (1:200) |
| Antibody | Mouse monoclonal anti-mouse PDL1 PE | Thermo Fisher Scientific | Cat# 12-5982-82 | FCS (1:200) |
| Antibody | Mouse monoclonal anti-mouse PDL2 PE | Thermo Fisher Scientific | Cat# 12-5986-82 | FCS (1:200) |
| Antibody | Mouse monoclonal anti-mouse IFNγ PE | Thermo Fisher Scientific | Cat# 12-7311-82 | FCS (1:100) |
| Antibody | Mouse monoclonal anti-mouse IL-4 PE | Thermo Fisher Scientific | Cat# 12-7041-82 | FCS (1:100) |
| Antibody | Rat monoclonal anti-mouse Foxp3 PE | Thermo Fisher Scientific | Cat# 12-5773-82 | FCS (1:100) |
| Antibody | Rat monoclonal anti-mouse IL-12p40 | Thermo Fisher Scientific | Cat# 12-7123-82 | FCS (1:100) |
| Antibody | Rat monoclonal anti-mouse TGFβ1 | Thermo Fisher Scientific | Cat# 12-9829-42 | FCS (1:100) |
| Commercial assay or kit | Fixation/Permeabilization Solution Kit | BD Bioscience | Cat# 554714 | |
| Commercial assay or kit | RNeasy Mini Kit | Qiagen | Cat# 74106 | |
| Commercial assay or kit | FastQuant RT Kit | Tiangen | Cat# KR106-02 | |
| Commercial assay or kit | Mouse IL-12 p70 Quantikine ELISA Kit | R&D Systems | Cat# M1270 | |
| Commercial assay or kit | Mouse TGF-beta 1 DuoSet ELISA | R&D Systems | Cat# DY1679 | |
| Commercial assay or kit | SuperReal PreMix Plus SYBR Green | Tiangen | Cat# FP205-02 | |
| Chemical compound, drug | Cyclosporin A | Signa-Aldrich | Cat# 59865-13-3 | |
| Chemical compound, drug | Yoda1 | MCE | Cat# HY-18723 | |
| Chemical compound, drug | 11R-VIVIT | MCE | Cat# HY-P1430 | |
| Chemical compound, drug | Gadolinium chloride | Sigma-Aldrich | Cat# 19423-81-5 | |
| Chemical compound, drug | Ruthenium red | Sigma-Aldrich | Cat# 12790-48-6 | |
| Chemical compound, drug | 2-Deoxy-D-glucose | Sigma-Aldrich | Cat# 29702-43-0 | |
| Chemical compound, drug | Diethyl succinate | Sigma-Aldrich | Cat# 123-25-1 | |
| Chemical compound, drug | IL-4 | R&D Systems | Cat# 204-IL | |
| Chemical compound, drug | IL-12 | Peprotech | Cat# 210–12 | |

*Continued on next page*

*Continued*

| Reagent type (species) or resource | Designation | Source or reference | Identifiers | Additional information |
|---|---|---|---|---|
| Chemical compound, drug | Fibronectin | Sigma-Aldrich | Cat# ECM001 | |
| Chemical compound, drug | Carbonyl cyanide-4-[trifluoromethoxy] phenylhydrazone | Sigma-Aldrich | Cat# SML2959 | |
| Chemical compound, drug | Rotenone | Sigma-Aldrich | Cat# R8875 | |
| Chemical compound, drug | Fura2 AM | Sigma-Aldrich | Cat# 47989 | |
| Chemical compound, drug | LPS | Sigma-Aldrich | Cat# 2630 | |
| Chemical compound, drug | Freund's Adjuvant, Complete | Sigma–Aldrich | Cat# F5881 | |
| Chemical compound, drug | PMA | Sigma-ALdrich | Cat# P8139 | |
| Chemical compound, drug | Ionomycin | Sigma-Aldrich | Cat# I0634 | |
| Chemical compound, drug | GM-CSF | R&D Systems | Cat# 7954GM-010/CF | |
| Chemical compound, drug | Collagenase D | Worthington | Cat# LS005273 | |
| Chemical compound, drug | Deoxyribonuclease I | Beyotime Biotechnology | Cat# D7076 | |
| Chemical compound, drug | Percoll | GE Health | Cat# 17-0891-01 | |
| Software, algorithm | Adobe Illustrator | Adobe | RRID:SCR_010279 | |
| Other | One-micrometer latex microspheres | Polysciences, Inc | Cat# 19821 | |
| Other | 7-AAD | BD Bioscience | Cat# 559925 RRID: AB_2869266 | A dye for identifying cell death |
| Other | Streptavidin-APC/eFluor 780 | Thermo Fisher Scientific | Cat# 47-4317-82; RRID:AB_10366688 | A secondary antibody for indirect staining to detect biotinylated primary antibody; FCS (1:500) |
| Other | Streptavidin-eFluor 450 | Streptavidin-eFluor 450 | Cat# 48-4317-82; RRID:AB_10359737 | A secondary antibody for indirect staining to detect biotinylated primary antibody; FCS (1:500) |
| Other | SYLGARD 527 A&B Silicone Dielectric Gel | Dow | | Silicone Dielectric Gel |

## Mice

All animal experiments were approved by the Animal Ethics Committee of Fudan University, Shanghai, China, Beijing Institute of Microbiology and Epidemiology and Beijing Normal University (IACUC-DWZX-2017-003 and CLS-EAW-2017-002) Beijing, China. C57BL/6 *Piezo1*^flox/flox, and *Piezo1*^-/- mice were obtained from Jicuiyaokang company of China (Nanjing, China). *Cd11c-Cre* mice were obtained from Nanfangmoshi biological company of China (Shanghai, China). *Sirt1*^flox/flox and *Hif1a*^flox/flox mice were obtained from the Jackson Laboratory (Bar Harbor, ME). OTII TCR-transgenic mice were obtained from Jicuiyaokang company of China (Nanjing, China). CD45.1 mouse was obtained from Beijing University Experimental Animal Center (Beijing, China). C57BL/6 mice were obtained from Fudan University Experimental Animal Center or Beijing University Experimental Animal Center (Beijing, China). All the mice had been backcrossed to the C57BL/6 background for at least eight generations and were used at an age of 6–12 weeks. WT control mice were of the same genetic background and,

where relevant, included Cre$^+$ mice to account for the effects of Cre (no adverse effects due to Cre expression itself were observed in vitro or in vivo).

## Tumor model

To establish subcutaneous tumors, $5\times10^5$ MC38 or MC38-OVA tumor cells or $4\times10^5$ B16.F10 melanoma cells were injected into C57BL/6 mice, half male and half female, randomization group. These cells formed a tumor of 1–2 cm diameter within 2–4 weeks of injection and double blinding detection of mouse tumor size.

## T cell isolation from tumor

Tumor tissues were cut into pieces and suspended using collagenase D (Worthington, NJ; 400 U/ml) and deoxyribonuclease I (Beyotime Biotechnology, Shanghai, China; 4 U/ml) in 2 ml complete RPMI 1640 medium (Corning). After incubated for 30 min in a shaker at 37°C, the homogenized tissue was passed through a cell strainer (70 μm; BD Pharmingen). After centrifugation at 200× $g$ for 5 min, the supernatant was discarded, and the pellet was resuspended with 3 ml of serum-free RPMI 1640. Add 3 ml 70% Percoll (GE Healthcare) to 15 ml centrifuge tube, then gently add 3 ml 40% Percoll to it, and finally slowly add 3 ml cell suspension to it to form a complete interface between the three layers. Centrifugation at 400× $g$ for 25 min with slow acceleration and without breaks created a gradient. The interface cells between two density Perocoll (about 2–4 ml) were collected, washed with PBS for two times and stained with T cell antibodies for sorting by flow cytometry and prepared for further analysis.

## Cell isolation from gut-associated lymphatic tissues

Isolation of LPLs was performed as previously (*Li et al., 2017*). The small intestine and large intestine were removed, opened longitudinally, and cut into pieces. After vigorous shaking in HBSS containing EDTA, the supernatants containing epithelial cells and IELs was discarded. The remaining intestinal pieces were digested with collagenase D (Worthington) and pelleted. The pellet was resuspended and placed in a Percoll gradient as described above, and after centrifugation, the interface containing the LPLs was collected and prepared for further analysis.

## Cell adoptive transfer

Naïve T cells (CD4$^+$TCR$^+$CD62L$^{hi}$CD44$^{lo}$CD25$^-$) from C57BL/6 mice or OTII TCR-transgenic mice were sorted and transferred into recipient mice. After 24 hr, the recipient mice were injected s.c. with WT and *Piezo1*$^{-/-}$ DCs mixed with OVA$_{323-339}$ in the presence of complete Freund's adjuvant (CFA; Difco), LPS (Sigma), or 50 kPa hydrogel-conditioned DCs. At days 8–9 after immunization, DLN cells were harvested and stimulated with their cognate peptides for 2–3 days prior to cytokine mRNA expression and secretion analyses or pulsed with PMA-ionomycin for 5 hr prior to the intracellular staining of donor-derived T cells.

## Cell cultures and flow cytometry

Spleens were digested with collagenase D, and DCs (CD11c$^+$TCR$^-$CD19$^-$NK1.1$^-$F4/80$^-$Ly6G$^-$) were sorted with a FACSAria II (Becton Dickinson, San Diego, CA). Naïve T cells were sorted from spleen or PLN. For DC-T cell cocultures, DCs and T cells (1:10) were mixed in the presence of 1 μg/ml OVA$_{323-339}$ peptide and 100 ng/ml LPS. After 5 days of culture, live T cells were stimulated with PMA and ionomycin for intracellular cytokine staining or with plate-bound α-CD3 to measure cytokine secretion and mRNA expression. T cell proliferation was determined by pulsing cells with $^3$H-thymidine for the final 12–16 hr of culture, as previously described (*Liu et al., 2009*). For drug treatments, the cells were incubated with vehicle, CsA (10 nM; Sigma), Yoda1 (25 μM, MCE), 11R-VIVIT (100 nM, MCE), ruthenium red (30 μM, Sigma), gadolinium chloride (10 μM, Sigma), 2-DG (1 mmol/l, Sigma), or diethyl succinate (1 mmol/l, Sigma) for 0.5–1 hr before stimulation. For antibody or cytokine treatment, cultures were supplemented with IL-12 (10 μg/ml, Peprotech) and anti-TGFβ1 mAb (20 μg/ml, R&D Systems). Flow cytometry was performed with the following antibodies from eBioscience, BD Biosciences, or Abcam: anti-CD11c FITC (N418), anti-CD11c PE (N418), anti-CD11c FITC (N418), anti-CD4 APC-Cy7 (GK1.5; Cat#130-109-536, RRID:AB_2657974), anti-CD8α FITC (53–6.7), anti-CD11b FITC (M1/70), anti-Ly6G PE (RB6-8C5; Cat# ab25378, RRID:AB_470493), anti-F4/80 PE (BM8), anti-CD19 PE (1D3; Cat#340418,

RRID:AB_400423), anti-TCR FITC (H57-597), anti-CD44 FITC (IM7), anti-CD62L APC (MEL14), anti-CD80 APC (1C10), anti-CD54 FITC (YN1/1/7.4), anti-MHCII (AF6-120), anti-CD45 APC (30-F11), anti-NK1.1 PE (PK136), anti-CCR7 APC (4B12, Cat#A18389, RRID: AB_2535249), anti-PDL1 PE (MIH5, Cat# 12-5982-82), anti-PDL2 PE (TY25, Cat# 12-5986-82), anti-IFNγ PE (XMG1.2), anti-IL-4 PE (11B11), anti-Foxp3 PE (FJK-16s), anti-mouse IL-12p40 mAb (241812; Cat#BE0051, RRID:AB_1107698), and anti-mouse TGFβ1 mAb (EPR21143). Flow cytometry data were acquired on a FACSCalibur (Becton Dickinson, CA) and the data were analyzed with FlowJo (RRID:SCR_008520; Tree Star, San Carlos, CA).

## Phagocytosis assay

One-micrometer latex microspheres (Polysciences, Inc) were incubated overnight with 1 mg/ml of mouse IgG-FITC (Jackson ImmunoResearch Laboratories). After rinsing the unbound antibody, the microspheres were added to DCs cultured at a ratio between DC and beads is 1:10 and incubated for 30 min in a humidified incubator at 37°C and 5% $CO_2$. Unbound or attached noninternalized beads were washed in consecutive rinsing steps with cold PBS. DCs were then stained with anti-CD11c, rinsed in cold PBS. The phagocytosis percentage of DCs were determined with flow cytometry.

## Hydrogel-coated plates

Dow Corning Sylgard 527 (Parts A and B, Sigma-Aldrich) was used to prepare PDMS hydrogel-coated plates. Part A and Part B of the gel were mixed to achieve the appropriate tension, as described (*Solis et al., 2019*; *Chakraborty et al., 2021*; *Liu et al., 2018*). For the 2 kPa gel, the ratio of A:B was 1:2, and for the 50 kPa gel, the ratio of A:B was 0.3. The plates were coated with the hydrogel and incubated overnight at 60°C. Then, the gels were coated with fibronectin (1 µg/ml, Sigma) for 4 hr at 37°C and washed again with PBS.

## Oxygen consumption analysis

Cells were plated in 24-well Seahorse plates at $2×10^5$ cells per well, and a negative control well containing only media without cells was included. A utility plate containing calibrant solution (1 ml/well) together with the plates containing the injector ports and probes was incubated in a $CO_2$-free incubator at 37°C overnight. The following day, the medium was removed from the cells and replaced with glucose-supplemented XF assay buffer (500 µl/well), and the cell culture plate was incubated in a $CO_2$-free incubator for at least 0.5 hr. Inhibitors (oligomycin, carbonyl cyanide-4-[trifluoromethoy]) phenylhydrazone, 2-DG, and rotenone (70 µl) were added to the appropriate port of the injector plate. This plate, together with the utility plate, was run on the Seahorse for calibration. Then, the utility plate was replaced with the cell culture plate, and the cell culture plate was analyzed on the Seahorse XF-24 instrument.

## Measurement of intracellular $Ca^{2+}$ concentrations

The intracellular $Ca^{2+}$ concentrations ($[Ca^{2+}]$) were measured fluorometrically using the fluorescent calcium indicator dye Fura2 AM (Sigma), as previously described (*Atcha et al., 2021*). Cells were incubated with 5 µM Fura2 AM in HBSS supplemented with 110 mM NaCl, 5 mM KCl, 0.3 mM $Na_2HPO_4$, 0.4 M $KH_2PO_4$, 5.6 mM glucose, 0.8 mM $MgSO_4$, 7 mM $H_2O$, 4 mM $NaHCO_3$, 1.26 mM $CaCl_2$, and 15 mM HEPE, at pH 7.4, at room temperature for 60 min.

## RNA and protein expression analysis

RNA was extracted with a RNeasy kit (QIAGEN, Dusseldorf, Germany), and cDNA was synthesized using SuperScript III reverse transcriptase (Invitrogen, Carlsbad, CA). An ABI 7900 real-time PCR system was used for quantitative PCR, with primer and probe sets obtained from Applied Biosystems (Carlsbad, CA). The results were analyzed using SDS 2.1 software (Applied Biosystems). The cycling threshold value of the endogenous control gene (*Hprt*1, which encodes hypoxanthine guanine phosphoribosyl transferase) was subtracted from the cycling threshold ($\Delta C_T$). The expression of each target gene is presented as the fold change relative to that of control samples ($2^{-\Delta\Delta CT}$). For the detection of phosphorylated signaling proteins, purified cells were activated with LPS (Sigma), immediately fixed with Phosflow Perm buffer (BD Biosciences) and stained with phycoerythrin or allophycocyanin directly conjugated to antibodies against Erk phosphorylated at Thr202 and Tyr204 (20A; Cat# 612566, RRID:AB_399857; BD Biosciences), p38MAPK phosphorylated at Thr180 and Thr182 (D3F9;

Cell Signaling Technology), JNK phosphorylated at Thr183 and Tyr185 (G9; Cell Signaling Technology), STAT4 phosphorylated at Tyr701 and Ser727 (58D6; Cell Signaling Technology), and SMAD3 phosphorylated at Tyr705 and Ser727 (D3A7; Cell Signaling Technology), as described (*Liu et al., 2009*). Intracellular staining analysis was performed as described (*Li et al., 2017*) using anti-HIF-1α (EPR16897; Abcam) and anti-SIRT1 (19A7AB4; Abcam) antibodies.

### IL-12Rβ2 and TGFβR2 knockdown with RNAi

A gene-knockdown lentiviral construct was generated by subcloning gene-specific short hairpin RNA (shRNA) sequences into lentiviral shRNA expression plasmids (pMagic4.1) as described (*Wang et al., 2016*). Lentiviruses were harvested from the culture supernatant of 293T cells (KCB Cat# KCB 200744YJ, RRID: CVCL_0063) transfected with shRNA vector. Sorted OTII CD4+ T cells were infected with the recombinant lentivirus, and green fluorescent protein-expressing cells were isolated using fluorescence sorting 48 hr later. IL-12Rβ2 and TGFβR2 expression was confirmed using real-time PCR. The sorted T cells expressing either control or shRNA vectors were used for functional assays.

### Human DC and T cell cultures

For assays of human DC-mediated T cell differentiation, normal human DCs (CC-2701; Lonza) were cultured and their populations were expanded for 5 days with human granulocyte-macrophage colony-stimulating factor and IL-4 (R&D Systems), followed by treatment with Yoda1 (25 μM, MCE) and stimulation for 24 hr with LPS. DCs were washed extensively and cultured with human cord blood CD4+ T cells (2C-200; Lonza) at a ratio of 1:10. After 7 days of culture, live T cells were purified and then stimulated either with PMA and ionomycin for intracellular cytokine staining for 5 hr or with plate-bound anti-CD3 for analysis of mRNA expression.

### Statistical analysis

All the data are presented as the mean ± s.d. Student's unpaired $t$ test was used for the comparison of means to evaluate differences between groups. A p value (alpha-value) of less than 0.05 was considered statistically significant.

### Acknowledgements

The authors' research is supported by grants from the National Natural Science Foundation for Key Programs of China (31730024, GL), National Natural Science Foundation for General Programs of China (32170911, GL and 31970863, YB), and Beijing Municipal Natural Science Foundation of China (5202013, GL).

## Additional information

### Funding

| Funder | Grant reference number | Author |
| --- | --- | --- |
| National Natural Science Foundation for Key Programme of China | 31730024 | Guangwei Liu |
| National Natural Science Foundation for General Program of China | 32170911 | Guangwei Liu |
| Beijing Municipal Natural Science Foundation | 5202013 | Guangwei Liu |
| National Natural Science Foundation for General Program of China | 31970863 | Yujing Bi |

The funders had no role in study design, data collection and interpretation, or the decision to submit the work for publication.

## Author contributions
Yuexin Wang, Data curation, Formal analysis, Investigation; Hui Yang, Resources, Data curation, Investigation, Project administration; Anna Jia, Data curation, Investigation; Yufei Wang, Participating in discussion; Qiuli Yang, Participating in discussion; Yingjie Dong, Data curation, Methodology; Yueru Hou, Participating in discussion; Yejin Cao, Participating in discussion; Lin Dong, Participating in discussion; Yujing Bi, Resources, Data curation, Supervision, Investigation; Guangwei Liu, Conceptualization, Resources, Data curation, Formal analysis, Supervision, Funding acquisition, Investigation, Writing - original draft, Project administration, Writing - review and editing

## Author ORCIDs
Guangwei Liu (iD) http://orcid.org/0000-0002-6008-2891

## Ethics
Normal human DCs (CC-2701; Lonza) and human cord blood CD4$^+$ T cells (2C-200; Lonza) were obtained from Lonza Company. All human subject experiments were performed with the approval of the Ethics Committee of of Fudan University, China and Beijing Normal University, China.

All animal experiments were approved by the Animal Ethics Committee of Fudan University, Shanghai, China, Beijing Institute of Microbiology and Epidemiology and Beijing Normal University (IACUC-DWZX-2017-003 and CLS-EAW-2017-002).

## Decision letter and Author response
Decision letter https://doi.org/10.7554/eLife.79957.sa1
Author response https://doi.org/10.7554/eLife.79957.sa2

---

# Additional files

## Supplementary files
• MDAR checklist

## Data availability
All data generated or analysed during this study are included in the manuscript and supporting files; source data files have been provided for Figure1.

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
