## [Editor Report]

In the present study, the authors use mouse cancer models to study the role of Piezo1 on DC-mediated priming of CD4^+^ T cells. They show that Piezo1 knockout results in faster tumor progression and accumulation of more regulatory T cells, and that Smad3 and STAT4 are involved in DC-mediated differentiation of T_H_1 and T_reg_ cells. Overall this represents a mechanistic advance in our understanding of DC biology as it relates to cancer. This also has the potential to extend beyond cancer to better our understanding of DC-mediated T cell differentiation.

---

## [Decision Letter]

**Decision letter after peer review:**

Thank you for submitting your article "Dendritic cell Piezo1 integrating mechanical stiffness and inflammatory signals directs the differentiation of T_H_1 and T_reg_ cells in cancer" for consideration by *eLife*. Your article has been reviewed by 3 peer reviewers, including Kellie N Smith as Reviewing Editor and Reviewer #1, and the evaluation has been overseen by Tadatsugu Taniguchi as the Senior Editor. The following individual involved in the review of your submission has agreed to reveal their identity: Walter Storkus (Reviewer #3).

Essential revisions:

1) Exploring the expression of Piezo1 in human DC.

2) Changes to the narration, text, and length as recommended by the reviewers.

3) TGFb Ab and recombinant IL-12 rescue experiments.

*Reviewer #1 (Recommendations for the authors):*

1. Overall, the study is too long and could benefit from a focus on the key novel findings. While this reviewer appreciates the several instances of orthogonal validation of the salient findings, less text "real estate" could be devoted to these confirmatory experiments.

2. Most of Fig. 3 could be removed or shortened.

3. Line 135: "Thus, we conclude that DC-specific Piezo1 deficiency alters the differentiation of Th1 and Treg cells during age-related T cell responses." -The data do not show that the immunological effects are age-related, but rather show that the clinical manifestation is age-related.

*Reviewer #2 (Recommendations for the authors):*

The key Th1, Th2, and Th17 transcription factors should be included in the work.

In Figure 1, the authors showed enhanced tumor growth in Piezo1ΔDC mice compared to WT. They showed increased FoxP3+ Tregs and decreased IFNγ+ Th1 cells in tumors of Piezo1ΔDC mice. They suggested these CD4 T cell changes probably promoted the tumor growth. This is not conclusive at all. How about CD8 T cell? Are there any changes in CD8 T cell number and function between Piezo1ΔDC and WT mice?

In Figure 4, the authors claimed that DC Piezo1 regulated T cell differentiation through IL-12 and TGFb1 signaling in vitro. Can they confirm these findings in vivo? Can anti-TGFb antibody or recombinant IL-12 protein reverse tumor phenotypes of Piezo1ΔDC mice?

The authors mentioned that there are no differences in the phenotype (CD80, CD86, MHC II) of DCs between WT and Piezo1ΔDC mice. How about key cytokine production or phagocytic activity/Ag presentation capacity?

All of the experiments were performed in mouse cells and one single mouse tumor model (MC38). Do human dendritic cells express Piezo1? Could they show comparable results in another tumor model?

*Reviewer #3 (Recommendations for the authors):*

Technical concerns and specific changes in text/figures were suggested to the authors for improvement.

1) IL-12p40 was analyzed for Type-1 DC output, yet IL-12p40 can homodimerize with itself or heterodimerize with IL-12p35 or IL-23p19 to produce either IL-12p70 or IL-23, respectively. The authors should indicate if Piezo1-activated DC produce more IL-12p40 homodimers, IL-12p70, and/or IL-23 to further refine their reported data set.

2) To better balance the T cell output cytokines associated with functional polarization, it is recommended that the authors include a Treg-kine, such as IL-10. Th1 (IFNG), Th2 (IL-4), and Th17 (IL-17) were used in the report, but Treg are only marked by Foxp3 expression.

3) Within the context of DC phenotype profiling, the authors suggest that Piezo1 status does not modulate DC expression of CD54, CD80 or CD86, suggesting a lack of impact on DC maturation. The authors should report whether Piezo1(deltaDC) expresses lower levels of CCR7, higher levels of checkpoint molecules (i.e. PD-L1, PD-L2, etc.), and higher rates of apoptosis vs. wild-type DC which could also factor into whether Type-1 vs. Treg responses are stimulated by these APCs.

4) Does intratumoral delivery of Yoda1 promote MC38 tumor growth control in wild-type mice in association with Type 1 >> Treg TIL phenotype?

5) At the end of the Discussion, the authors state "(these findings have)...implications for targeting DCs as an approach to the treatment of cancer". This should be a bit more forward of a statement. How do the authors envision these data informing future clinical trial design in the cancer setting? (application of Piezo1 agonists such as Yoda1, etc.). Given a range of alternate DC agonists that would be envisioned to potentiate Type-1 T cell responses (i.e. TLR agonists, STING agonists, among others), please invest a bit more effort in developing a translational paradigm here that would speak to the preferred use of Piezo1-targeted strategies.

6) In Fig. 1, the faster growth rate of MC38 tumors in Piezo1(deltaDC) mice is presumed to be due to more Treg and fewer Type-1 (CD8+ T cells), but data are merely correlative. Can the authors speak to whether depletion of Treg in the KO mice results in control tumor growth (as observed in wild-type mice)?

7) For the studies involving T cell coculture with (2 kPa vs. 50 kPa) hydrogel-conditioned DCs + Antigen, are the DCs isolated first and then cocultured with T cells, or are T cells merely added to the DC-conditioned cultures? If the latter, the authors should comment on intrinsic stiffness-related aspects in the T cell responders as it relates to study outcomes.

8) Fig. 4A right (MFI) panels require labels for TGFB1 and IL-12.

9) The title for Supplemental Figure 6 is incorrect as it reports comparative expression of TGFBR1, TGFBR3, and IL-12RB1 on DC. The title for Supplemental Figure 7 is incorrect as it merely shows the impact of shRNA knockdown on target gene expression in DC. Please correct.

10) Similarly, the title for Supplemental Figures 8, 9, and 11/12/14 appear incorrect since TGFB1/IL-12 are not readouts in the data presented in Fig. S8, SIRT1/HIF1A are not interrogated in Fig. S9 and IL-12 is not an outcome index in Fig. S11/12/14. Please correct.

11) In the cartoon supplied in Supplemental Figure 15, the bottom "Promoting cancer growth" should be deleted and replaced by a red legend for "Tumor progression" below the Treg column, and a yellow legend for Tumor Growth Control" under the Th1 column for reader clarity.

12) Although generally well-written, the report is very long and should be consolidated as much as is possible.

---

## [Author Response]

Essential Revisions (for the authors):1) Exploring the expression of Piezo1 in human DC.

Thanks for the editor and reviewer’s suggestions. We have included the new data in Figure 7C-H in the revised manuscript.

2) Changes to the narration, text, and length as recommended by the reviewers.

Thanks for the editor and reviewer’s suggestions. Following the reviewer’s comments, we have revised the manuscript now.

3) TGFb Ab and recombinant IL-12 rescue experiments.

Thanks for the editor and reviewer’s comments. We have included the new data in Figure 7A-B to show that changing IL-12 and TGFβ signaling can significantly alter T cell differentiation in the tumor environment in anti-tumor immunity in revised manuscript.

Reviewer #1 (Recommendations for the authors):1. Overall, the study is too long and could benefit from a focus on the key novel findings. While this reviewer appreciates the several instances of orthogonal validation of the salient findings, less text "real estate" could be devoted to these confirmatory experiments.

Thanks for the reviewer’s suggestions. Following the reviewer’s comments, we have revised and shortened the text.

2. Most of Fig. 3 could be removed or shortened.

Thanks. Following the reviewer’s comments, we have removed the original Fig.3 to Figure 3-figure supplement 1-2 now.

3. Line 135: "Thus, we conclude that DC-specific Piezo1 deficiency alters the differentiation of Th1 and Treg cells during age-related T cell responses." -The data do not show that the immunological effects are age-related, but rather show that the clinical manifestation is age-related.

Thanks. Following the reviewer’s suggestions, we have revised the description in the revised manuscript.

Reviewer #2 (Recommendations for the authors):The key Th1, Th2, and Th17 transcription factors should be included in the work.

Thanks. Following the reviewer’s suggestions, we have included the new data in Figure 1—figure supplement 4C, Figure 3—figure supplement 2A, and also added the descriptions accordingly in the revised manuscript.

In Figure 1, the authors showed enhanced tumor growth in Piezo1ΔDC mice compared to WT. They showed increased FoxP3+ Tregs and decreased IFNγ+ Th1 cells in tumors of Piezo1ΔDC mice. They suggested these CD4 T cell changes probably promoted the tumor growth. This is not conclusive at all. How about CD8 T cell? Are there any changes in CD8 T cell number and function between Piezo1ΔDC and WT mice?

Thanks for the reviewer’s suggestions and comments. Following the reviewer’s suggestions, we revised the text and have also included the new data in Figure 1—figure supplement 4A-B to show the CD8^+^T cell number and IFNγ production in tumor in WT and *Piezo1*^-/-^ mice and added the new descriptions in the revised manuscript.

In Figure 4, the authors claimed that DC Piezo1 regulated T cell differentiation through IL-12 and TGFb1 signaling in vitro. Can they confirm these findings in vivo? Can anti-TGFb antibody or recombinant IL-12 protein reverse tumor phenotypes of Piezo1ΔDC mice?

Thanks for the reviewer’s comments. We have included the new data in Figure 7A-B to show that changing IL-12 and TGFβ signaling can significantly reverse the tumor growth and alter T cell differentiation in the tumor environment in revised manuscript.

The authors mentioned that there are no differences in the phenotype (CD80, CD86, MHC II) of DCs between WT and Piezo1ΔDC mice. How about key cytokine production or phagocytic activity/Ag presentation capacity?

Thanks for the reviewer’s suggestions. We have included the new data in Figure 3—figure supplement 4A-C, Figure 3—figure supplement 3D and added the descriptions about the Figure 2A and Figure 2D in the revised manuscript.

All of the experiments were performed in mouse cells and one single mouse tumor model (MC38). Do human dendritic cells express Piezo1? Could they show comparable results in another tumor model?

Thanks for the reviewer’s suggestions. We have included the new data in Figure 1—figure supplement 5A-B (B16.F10 melanoma tumor) and Figure 7C-H (human DCs) in the revised manuscript.

Reviewer #3 (Recommendations for the authors):Technical concerns and specific changes in text/figures were suggested to the authors for improvement.1) IL-12p40 was analyzed for Type-1 DC output, yet IL-12p40 can homodimerize with itself or heterodimerize with IL-12p35 or IL-23p19 to produce either IL-12p70 or IL-23, respectively. The authors should indicate if Piezo1-activated DC produce more IL-12p40 homodimers, IL-12p70, and/or IL-23 to further refine their reported data set.

Thanks for the reviewer’s suggestions. We have included the new data in Figure 3-figure supplement 4B-C in the revised manuscript.

2) To better balance the T cell output cytokines associated with functional polarization, it is recommended that the authors include a Treg-kine, such as IL-10. Th1 (IFNG), Th2 (IL-4), and Th17 (IL-17) were used in the report, but Treg are only marked by Foxp3 expression.

Thanks for the reviewer’s suggestions. We have included the new data in Figure 1-figure supplement 4C and Figure 3-figure supplement 2A in the revised manuscript.

3) Within the context of DC phenotype profiling, the authors suggest that Piezo1 status does not modulate DC expression of CD54, CD80 or CD86, suggesting a lack of impact on DC maturation. The authors should report whether Piezo1(deltaDC) expresses lower levels of CCR7, higher levels of checkpoint molecules (i.e. PD-L1, PD-L2, etc.), and higher rates of apoptosis vs. wild-type DC which could also factor into whether Type-1 vs. Treg responses are stimulated by these APCs.

Thanks for the reviewer’s suggestions. We have included the new data in Figure 3-figure supplement 3C and 3A in the revised manuscript.

4) Does intratumoral delivery of Yoda1 promote MC38 tumor growth control in wild-type mice in association with Type 1 >> Treg TIL phenotype?

Thanks for the reviewer’s suggestions. We have added discussion in the revised manuscript.

5) At the end of the Discussion, the authors state "(these findings have)...implications for targeting DCs as an approach to the treatment of cancer". This should be a bit more forward of a statement. How do the authors envision these data informing future clinical trial design in the cancer setting? (application of Piezo1 agonists such as Yoda1, etc.). Given a range of alternate DC agonists that would be envisioned to potentiate Type-1 T cell responses (i.e. TLR agonists, STING agonists, among others), please invest a bit more effort in developing a translational paradigm here that would speak to the preferred use of Piezo1-targeted strategies.

Thanks for the reviewer’s suggestions. Following the reviewer’s comments, we have revised the descriptions in the revised manuscript.

6) In Fig. 1, the faster growth rate of MC38 tumors in Piezo1(deltaDC) mice is presumed to be due to more Treg and fewer Type-1 (CD8+ T cells), but data are merely correlative. Can the authors speak to whether depletion of Treg in the KO mice results in control tumor growth (as observed in wild-type mice)?

Thanks for the reviewer’s suggestions. Following the reviewer’s comments, we have included the data in new Fig.7A-B and added the descriptions to show the T cell differentiation is critical for the tumor growth control induced by DC Piezo1 deficient.

7) For the studies involving T cell coculture with (2 kPa vs. 50 kPa) hydrogel-conditioned DCs + Antigen, are the DCs isolated first and then cocultured with T cells, or are T cells merely added to the DC-conditioned cultures? If the latter, the authors should comment on intrinsic stiffness-related aspects in the T cell responders as it relates to study outcomes.

Thanks for the reviewer’s suggestions. We isolated DCs first and then cocultured with T cells. Following the reviewer’s suggestions, we have revised the figure legend in the revised manuscript.

8) Fig. 4A right (MFI) panels require labels for TGFB1 and IL-12.9) The title for Supplemental Figure 6 is incorrect as it reports comparative expression of TGFBR1, TGFBR3, and IL-12RB1 on DC. The title for Supplemental Figure 7 is incorrect as it merely shows the impact of shRNA knockdown on target gene expression in DC. Please correct.10) Similarly, the title for Supplemental Figures 8, 9, and 11/12/14 appear incorrect since TGFB1/IL-12 are not readouts in the data presented in Fig. S8, SIRT1/HIF1A are not interrogated in Fig. S9 and IL-12 is not an outcome index in Fig. S11/12/14. Please correct.

Thanks for the reviewer’s criticisms and comments. We have corrected them in the figure legends.

11) In the cartoon supplied in Supplemental Figure 15, the bottom "Promoting cancer growth" should be deleted and replaced by a red legend for "Tumor progression" below the Treg column, and a yellow legend for Tumor Growth Control" under the Th1 column for reader clarity.

Thanks for the reviewer’s suggestions. Following the reviewer’s suggestions, we have revised the cartoon in the new Figure 7-figure supplement 1.

12) Although generally well-written, the report is very long and should be consolidated as much as is possible.

Thanks for the reviewer’s suggestions. Following the reviewer’s suggestions, we consolidated some contents and shortened the length of the article now.